# Learning Adaptive Perturbation-Conditioned Contexts for Robust Transcriptional Response Prediction

Yinhua Piao [1]  Hyomin Kim [2]  Seonghwan Kim [1]  Yunhak Oh [3]  Junhyeok Jeon [4]  Sang-Yeon Hwang [4]
Jaechang Lim [4]  Woo Youn Kim [1 5 4]  Chanyoung Park [6]  Sungsoo Ahn [1 2]

## Abstract

Predicting high-dimensional transcriptional responses to genetic perturbations is challenging because signals are sparse and experimental noise is severe. Existing methods often suffer from *mean collapse*, achieving high correlation by predicting the global average expression rather than perturbation-specific responses, which yields false positives and poor interpretability. Methods that add biological knowledge graphs typically treat them as dense, static priors shared across perturbations, propagating noise. We propose ADAPERT, which counters mean collapse by extracting a sparse, perturbation-specific subgraph via differentiable node selection, then suppressing spurious variation in non-responsive genes while emphasizing differentially expressed ones. Across multiple benchmarks, ADAPERT outperforms existing baselines, with the largest gains on DEG-aware metrics.

*Figure 1.* Mean-collapse as a common failure mode in perturbation modeling. Each point is a gene for the UQCRB perturbation, showing predicted vs. ground-truth expression change (Real Delta); gray points are all observed genes, colored points are the $n=114$ DEGs. **(a)** A standard model (Wenkel et al., 2025) exhibits mean-collapse: predictions for large-effect DEGs shrink toward zero, inflating overall correlation while underestimating true effects. **(b)** ADAPERT uses perturbation-specific context to better track DEG changes, raising DEG correlation from 0.617 to 0.689 (+11%).

## 1. Introduction

Predicting how cells respond to perturbations is a key problem in functional genomics (Shalem et al., 2015; Przybyla & Gilbert, 2022). It supports downstream tasks including understanding gene function (Kim et al., 2024), analyzing regulatory effects (Ishikawa et al., 2023; Yao et al., 2024), identifying therapeutic targets (Gonzalez et al., 2025; Kim et al., 2021; Shin et al., 2022). Recent progress in single-cell perturbation experiments, including Perturb-seq

and CRISPR-based screens (Datlinger et al., 2017; Bock et al., 2022), now allows gene expression to be measured across thousands of genes under many perturbation conditions (Dixit et al., 2016; Norman et al., 2019; Replogle et al., 2022). As a result, there is growing interest in computational models that can predict transcriptional responses to perturbations that have not been experimentally tested.

A central challenge in this task is that single-cell measurements are inherently noisy, making it difficult to learn perturbation-specific effects from the samples (Brennecke et al., 2013). Recent efforts have addressed this challenge primarily by enriching the input: scaling up training data (Cui et al., 2024), adding features with biological annotations (Chen & Zou, 2024), and incorporating knowledge graphs to improve generalization to unseen perturbations (Roohani et al., 2024). While these directions have shown progress, we observe that a fundamental failure mode persists across many existing methods. Instead of capturing perturbation-specific changes, models tend to predict ex-

---

[1]AI Co-Research & Education for innovative Drug Institute, KAIST [2]Kim Jaechul Graduate School of Artificial Intelligence, KAIST [3]Graduate School of Data Science, KAIST [4]HITS, Seoul, South Korea [5]Department of Chemistry, KAIST, Seoul, South Korea [6]Department of Industrial and Systems Engineering, KAIST, Seoul, South Korea. Correspondence to: Sungsoo Ahn <sungsoo.ahn@kaist.ac.kr>.

pression shifts close to the global average: a behavior we call **mean-collapse**. As illustrated in Figure 1(a), baseline model (Wenkel et al., 2025) collapses predictions toward the center of the distribution, where non-DEG genes dominate (*Gray* dots). This can produce high overall correlation, but expression changes of biologically important genes (the DEGs, shown as *Blue* dots) are strongly underestimated, which aligns with the recent findings (Mejia et al., 2025). As a result, these models yield many false positives and provide limited insight into the true effects of perturbations.

We trace mean collapse to how perturbation effects are modeled and supervised, rather than to data limitations. For each perturbation, only a small subset of genes—often *1–10% of measured genes*—shows strong expression changes, and these genes are typically direct interaction partners or downstream targets of the perturbed gene in known regulatory pathways (Wu et al., 2009; Replogle et al., 2022). Addressing this requires two things: separating perturbation-specific signal from background variation, and using biological structure to guide that separation. Existing knowledge-graph methods address neither directly (Wenkel et al., 2025; He et al., 2025). They use the graph as a dense, static input for global embedding rather than as a mechanism for selecting perturbation-relevant genes. As a result, signal spreads across unrelated genes and false positives accumulate.

Based on these observations, we propose ADAPERT, a perturbation-conditioned method that directly addresses mean-collapse by modeling sparsity and biological structure. Rather than treating knowledge graphs as fixed templates, ADAPERT learns a perturbation-conditioned context for each perturbation. Starting from control cells, the method selects genes related to the perturbed gene and extracts a compact subgraph from the full graph. An adaptive learning scheme then controls variation in non-responsive genes and uses responsive genes to refine the subgraph extraction. This design enables robust modeling of perturbation-specific transcriptional changes under the noisy settings.

We evaluate ADAPERT on four single-cell CRISPR perturbation benchmarks (K562, RPE1, HEPG2, and JURKAT) under the challenging *unseen-perturbation* setting. Across datasets, ADAPERT consistently outperforms these methods, with the largest gains on the DEG-aware metrics that are most sensitive to perturbation-specific signals. We further provide a comprehensive analysis stratified by perturbation effect-size, showing that ADAPERT's advantage persists even for small-effect perturbations — the regime where mean-collapse is most severe — improving differential-expression recovery by up to 28% over the strongest baseline. These results show that learning adaptive perturbation-conditioned context on biological knowledge graphs improves perturbation prediction in noisy settings.

## 2. Related Works

### 2.1. Data-Driven Genetic Perturbation Modeling

Genetic perturbation modeling has been extensively studied through data-driven learning approaches, which aim to reconstruct gene expression responses under perturbation conditions (Lopez et al., 2018; Lotfollahi et al., 2019; Bunne et al., 2023; Lotfollahi et al., 2023; Cui et al., 2024; Hao et al., 2024; Adduri et al., 2025). Most existing methods formulate this task as an end-to-end prediction problem, optimizing objectives such as mean squared error or correlation between predicted and observed expression profiles. These approaches have demonstrated strong performance on standard quantitative metrics and are widely adopted as baselines for perturbation prediction tasks. However, by primarily focusing on overall reconstruction accuracy, they do not explicitly model which genes are causally or specifically affected by a given perturbation, motivating the exploration of additional sources of inductive bias (Wenteler et al., 2024; Wu et al., 2024; Li et al., 2024).

### 2.2. Knowledge-Driven Genetic Perturbation Modeling

To address the limitations of purely data-driven approaches, recent work has explored incorporating prior knowledge to provide structural (Wenkel et al., 2025; Roohani et al., 2024; He et al., 2025) or semantic constraints (Cui et al., 2024; Chen & Zou, 2024; Istrate et al., 2024). Such priors aim to guide models toward biologically plausible solutions, improve robustness under noisy single-cell measurements, and better capture perturbation-specific regulatory effects. Existing approaches leverage structured biological resources, including curated networks and textual knowledge, but the integration of these priors is often static and global.

Biological knowledge graphs, such as protein–protein interaction networks and pathway databases, have been widely used to encode relationships among genes. In perturbation modeling, these graphs are typically incorporated to generate gene embeddings, constrain message passing, or regularize model parameters (Wenkel et al., 2025; Roohani et al., 2024; He et al., 2025). By propagating information along known biological interactions, graph-based methods introduce inductive biases that reflect prior biological knowledge. However, most existing approaches treat knowledge graphs as dense and static structures that are shared across all perturbations. As a result, the same global graph is applied regardless of the specific perturbation, without explicitly identifying which substructures are relevant to a given perturbation condition.

More recently, large language models (LLMs) have been explored as a means of extracting biological knowledge from unstructured text, including scientific literature and curated databases (Chen & Zou, 2024; Istrate et al., 2024). In biolog-

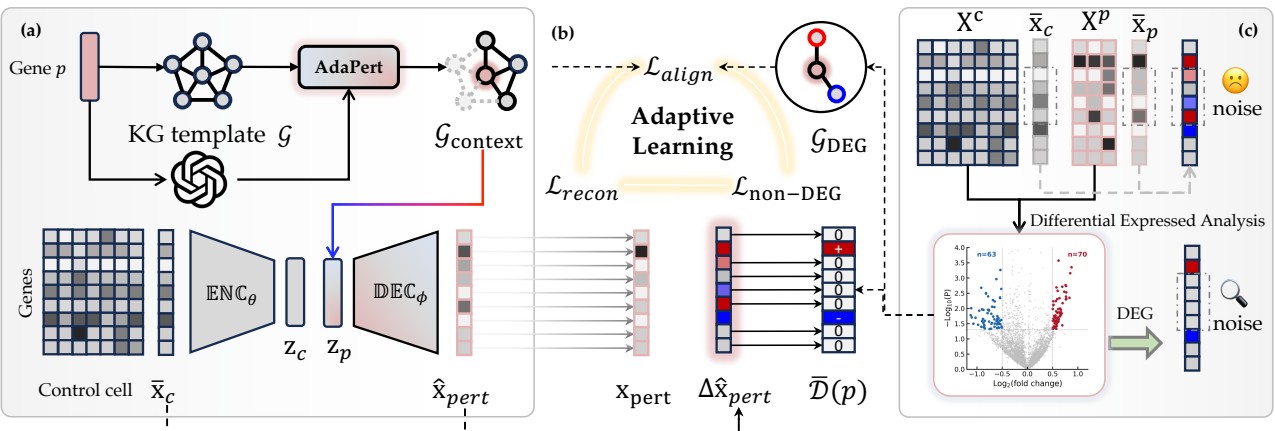

*Figure 2.* Overview of AdaPert (a) The model takes a control cell expression profile $\bar{\mathbf{x}}_c$ and a perturbation gene $p$ as input. A perturbation-conditioned subgraph $\mathcal{G}_{\text{context}}$ is extracted from a biological knowledge graph template $\mathcal{G}$, producing a context representation $\mathbf{z}_p$ that is combined with the encoded control state $\mathbf{z}_c$ to predict the perturbed expression profile $\hat{\mathbf{x}}_{\text{pert}}$ via encoder $\text{ENC}_\theta$ and decoder $\text{DEC}_\phi$. (b) The adaptive learning scheme separates signal from noise using three loss terms: a reconstruction loss $\mathcal{L}_{\text{recon}}$ for overall expression fidelity, a non-DEG loss $\mathcal{L}_{\text{non-DEG}}$ that suppresses spurious changes in non-responsive genes, and an alignment loss $\mathcal{L}_{\text{align}}$ that guides the learned subgraph representation $\mathcal{G}_{\text{DEG}}$ to encode perturbation-specific differential expression patterns. (c) Illustration of single-cell perturbation data structure showing control cells $\mathbf{X}^c$, perturbed cells $\mathbf{X}^p$, and their mean profiles $\bar{\mathbf{x}}_c$ and $\bar{\mathbf{x}}_p$. Differentially expressed genes (DEGs) are identified through statistical testing, distinguishing true perturbation signals from experimental noise.

ical applications, LLMs are commonly used for tasks such as gene annotation, relationship scoring, and semantic retrieval (Wu et al.; Istrate et al., 2025; He et al., 2025). These models provide a complementary source of prior knowledge that is difficult to encode in structured graphs alone. However, most LLM-based approaches do not directly model perturbation-response data and instead rely on inferred associations or reasoning over textual knowledge. Consequently, LLMs are often better suited as auxiliary components that provide prior guidance, rather than as standalone models for predicting perturbation-induced transcriptional responses.

## 3. Methodology

### 3.1. Problem Definition and Preliminaries

We formalize the task of predicting transcriptional responses to genetic perturbations within a conditional generative framework. Let $\mathbf{X}^c \in \mathbb{R}^N$ denote the gene expression profile of a control cell $c$, where $N$ is the number of observed genes. A perturbation targeting a specific gene $p$ is selected from the set $\mathcal{K}$. Our objective is to learn a predictive mapping $\mathcal{F} : (\mathbf{X}^c, p) \to \hat{\mathbf{X}}^p$, where $\hat{\mathbf{X}}^p \in \mathbb{R}^N$ is the predicted expression profile post-perturbation.

Existing state-of-the-art methods typically implement $\mathcal{F}$ using a conditional autoencoder backbone consisting of three functional components: a control encoder, a condition en-

coder, and a perturbation decoder. An encoder $\text{ENC}_\theta$ maps the control profile into a lower-dimensional latent representation $\mathbf{z}_c$, capturing the baseline state:

$$\mathbf{z}_c = \text{ENC}_\theta(\mathbf{X}^c), \quad \mathbf{z}_c \in \mathbb{R}^d \quad (1)$$

The perturbed gene $p$ (condition) is represented by an embedding $\mathbf{z}_p \in \mathbb{R}^d$. $\mathbf{z}_p$ is a learnable vector initialized by one-hot encoding or from recent gene foundation models (Cui et al., 2024; Theodoris et al., 2023). In more recent knowledge-guided models (Wenkel et al., 2025), it is derived from a global biological knowledge graph $\mathcal{G} = (\mathcal{V}, \mathcal{E})$ via a graph neural network (Veličković et al., 2018):

$$\mathbf{z}_p = \text{GNN}(\mathcal{G}, p) \quad (2)$$

where nodes $\mathcal{V}$ represent genes and edges $\mathcal{E}$ represent functional interactions. Finally, a decoder $\text{DEC}_\phi$ reconstructs the perturbed transcriptional response by integrating the cell state $\mathbf{z}_c$ and the perturbation signal $\mathbf{z}_p$:

$$\hat{\mathbf{X}}^p = \text{DEC}_\phi(\mathbf{z}_c, \mathbf{z}_p) \quad (3)$$

The model is trained by minimizing a reconstruction loss $\mathcal{L}(\mathbf{X}^p, \hat{\mathbf{X}}^p)$, defined as the mean squared error between the predicted and observed gene expression profiles.

However, this objective treats all genes equally. In genetic perturbation data, true responses are sparse, with only a

small subset of genes showing strong changes while most genes remain near baseline. Let $\mathcal{D}(p)$ denote the set of responsive genes (DEGs) under perturbation $p$, and $\bar{\mathcal{D}}(p)$ its complement, with $|\mathcal{D}(p)| \ll |\bar{\mathcal{D}}(p)|$. The reconstruction loss can be decomposed as

$$\mathcal{L}_{\text{rec}} = \underbrace{\sum_{i \in \mathcal{D}(p)} \left( \hat{\mathbf{X}}_i^p - \mathbf{X}_i^p \right)^2}_{\text{responsive genes}} + \underbrace{\sum_{i \in \bar{\mathcal{D}}(p)} \left( \hat{\mathbf{X}}_i^p - \mathbf{X}_i^p \right)^2}_{\text{non-responsive genes}}. \quad (4)$$

Since the second term dominates, minimizing mean squared error is driven mainly by non-responsive genes, encouraging predictions to shrink toward zero. As a result, large perturbation effects are systematically underestimated, leading to a failure mode we refer to as **mean-collapse**.

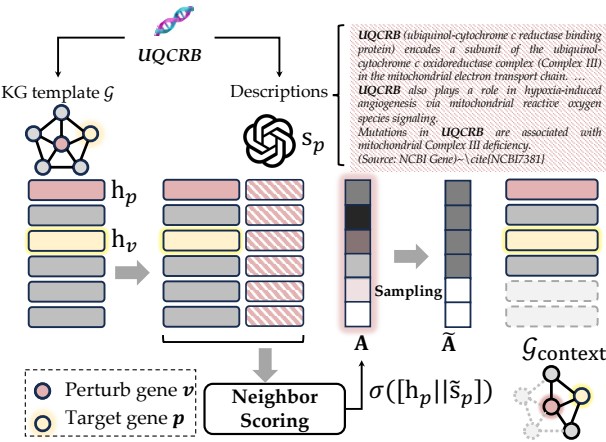

*Figure 3.* Perturbation-Conditioned Subraph Extraction. Given a perturbed gene $p$ (e.g., UQCRB), the module extracts a perturbation-specific subgraph from the knowledge graph template $\mathcal{G}$. First, a textual description of the perturbed gene is retrieved from NCBI and encoded using a language model to obtain a semantic embedding $\mathbf{s}_p$. Each node $v$ in the graph is represented by a structural embedding $\mathbf{h}_v$ computed via message passing. For neighbor scoring, the semantic embedding is concatenated with each node's structural embedding, and a perturbation-conditioned relevance score is computed via $\sigma([\mathbf{h}_v \| \tilde{\mathbf{s}}_p])$. Differentiable Gumbel-Softmax sampling is then applied to select a sparse set of perturbation-relevant nodes, yielding a compact subgraph $\mathcal{G}_{\text{context}}$ centered around genes related to the perturbed gene.

## 3.2. Overview of ADAPERT

To address the mean-collapse induced by dense reconstruction objectives, we propose ADAPERT, a perturbation-conditioned framework (Figure 2) that explicitly models sparsity of signals and biological structure related to the perturbation. The model consists of two components. **First**,

ADAPERT extracts a *perturbation-conditioned subgraph* from a unified biological knowledge graph template. Rather than using the full graph as a static prior, this module selects a compact subgraph that captures genes biologically related to the perturbed gene, providing a structured hypothesis space for perturbation response modeling. **Second**, ADAPERT employs an *adaptive learning* scheme to separate the true signal from noise. This module constrains spurious variations in non-differentially expressed genes while leveraging differentially expressed genes to guide the alignment and refinement of subgraph representations. Together, these two components enable perturbation-specific modeling that reduces noise propagation and preserves sparse transcriptional responses with high fidelity.

## 3.3. Perturbation-Conditioned Subgraph Extraction

We extract a perturbation-specific subgraph from a unified biological knowledge graph $\mathcal{G} = (\mathcal{V}, \mathcal{E})$. Instead of propagating messages over the full graph, our approach selects a sparse set of *perturbation-relevant* nodes to construct a subgraph centered around the perturbed gene. This design integrates semantic information beyond graph structure and reduces overfitting by restricting message passing to a small, condition-dependent context.

**Node Representations.** Each gene node $v \in \mathcal{V}$ is represented by a structural embedding that captures graph topology. We initialize each node with a one-hot vector $\mathbf{x}_v$ and apply message passing:

$$\mathbf{h}_v^{(0)} = \mathbf{x}_v, \quad (5)$$

$$\mathbf{h}_v^{(l+1)} = \sum_{u \in \mathcal{N}(v)} \frac{1}{|\mathcal{N}(v)|} \mathbf{W}^{(l)} \mathbf{h}_u^{(l)}, \quad (6)$$

where $\mathcal{N}(v)$ denotes the neighbors of $v$ and $\mathbf{W}^{(l)}$ are learnable weights. After $L$ layers, we obtain the structural embedding $\mathbf{h}_v = \mathbf{h}_v^{(L)} \in \mathbb{R}^{d_s}$.

**Perturbation Semantic Embedding.** We use language model embeddings to provide a basic semantic understanding of each perturbation gene. By encoding textual descriptions of the perturbed gene, the language model captures information that is not explicitly represented in the knowledge graph, such as gene family membership, functional similarity, and naming-related associations. This semantic information complements graph structure and enables the model to identify relevant genes that may be weakly connected or disconnected in the graph. For each perturbation gene $p$, we retrieve its textual description from NCBI and encode it using a language model (GPT-4o (OpenAI, 2024)):

$$\mathbf{s}_p = \text{LM}(\text{desc}(p)), \quad \mathbf{s}_p \in \mathbb{R}^{d_t}. \quad (7)$$

To align semantic and structural spaces, we project the perturbation embedding into the graph embedding space:

$$\tilde{\mathbf{s}}_p = \mathbf{W}_s \mathbf{s}_p, \tag{8}$$

where $\mathbf{W}_s \in \mathbb{R}^{d_s \times d_t}$.

**Perturbation-Conditioned Node Scoring.** To identify nodes relevant to a given perturbation, we condition node selection on the perturbation embedding. For each node $v$, we construct a joint representation by concatenating its structural embedding with the perturbation embedding:

$$\mathbf{c}_v = [\mathbf{h}_v \,\|\, \tilde{\mathbf{s}}_p]. \tag{9}$$

A perturbation-conditioned node relevance score is computed via a multilayer perceptron:

$$a_v = \mathbf{w}^\top \sigma(\mathbf{W}_c \mathbf{c}_v), \tag{10}$$

where $\sigma(\cdot)$ denotes a non-linear activation. Scores are normalized across all nodes:

$$\alpha_v = \frac{\exp(a_v)}{\sum_{u \in \mathcal{V}} \exp(a_u)}. \tag{11}$$

**Differentiable Node Sampling.** To enforce sparsity while preserving differentiability, we apply Gumbel-Softmax sampling (Jang et al., 2017) over node scores:

$$\tilde{\alpha}_v = \frac{\exp\left((\log \alpha_v + g_v)/\tau\right)}{\sum_{u \in \mathcal{V}} \exp\left((\log \alpha_u + g_u)/\tau\right)}, \tag{12}$$

where $g_v \sim \mathrm{Gumbel}(0, 1)$ and $\tau$ is a temperature parameter. Nodes with $\tilde{\alpha}_v > T$ are selected, yielding a perturbation-specific node-induced subgraph $\mathcal{G}_p$.

**Perturbation Context Representation.** Finally, we summarize the selected subgraph by aggregating the embeddings of the selected nodes:

$$\mathbf{z}_{\text{context}} = \sum_{v \in \mathcal{V}(\mathcal{G}_p)} \mathbf{h}_v. \tag{13}$$

Because node selection is explicitly conditioned on the perturbation, different perturbations induce distinct subgraphs, allowing the model to focus on causally relevant genes while filtering out unrelated graph structure.

### 3.4. Adaptive Learning for Signal–Noise Separation

We explicitly separate signal and noise during training by leveraging perturbation-specific differential expression information. For each perturbation $p$, let $\mathbf{X}^c, \mathbf{X}^p \in \mathbb{R}^N$ denote the control and perturbed expression profiles, and define the perturbation effect $\Delta \mathbf{X}^p = \mathbf{X}^p - \mathbf{X}^c$. Using the training data, we perform a statistical test for each gene and

obtain a $p$-value $q_i^{(p)}$. We define the DEG and non-DEG sets as

$$\mathcal{D}(p) = \{i \mid q_i^{(p)} < 0.05\}, \quad \bar{\mathcal{D}}(p) = \{i \mid q_i^{(p)} \geq 0.05\}. \tag{14}$$

Rather than treating all genes equally, we introduce three complementary loss terms with distinct roles: (i) a global reconstruction loss to preserve overall expression fidelity, (ii) a robust penalty that suppresses spurious changes on non-DEG genes, and (iii) a response-aware alignment loss that encourages the extracted subgraph representation to encode perturbation-specific DEG signals. Together, these objectives promote explicit separation between signal and noise during training.

**Global reconstruction loss.** We first match the full perturbed expression profile using a mean squared error:

$$\mathcal{L}_{\text{recon}} = \mathbb{E}\left[\|\hat{\mathbf{X}}^p - \mathbf{X}^p\|_2^2\right]. \tag{15}$$

This term ensures global consistency and stabilizes optimization, but alone is insufficient to distinguish true perturbation effects from noisy fluctuations.

**Non-DEG robust loss.** For non responsive genes $\bar{\mathcal{D}}(p)$, the expected perturbation change is close to zero, while experimental measurements can be noisy. To reduce spurious deviations without being overly sensitive to outliers, we penalize predicted perturbation changes on $\bar{\mathcal{D}}(p)$ using a Huber loss (Huber, 1992):

$$\mathcal{L}_{\text{non}} = \mathbb{E}_p\left[\sum_{i \in \bar{\mathcal{D}}(p)} \rho_\delta\left(\Delta \hat{\mathbf{X}}_i^p\right)\right], \qquad \Delta \hat{\mathbf{X}}^p = \hat{\mathbf{X}}^p - \mathbf{X}^c. \tag{16}$$

The Huber penalty is defined as

$$\rho_\delta(r) = \begin{cases} \frac{1}{2} r^2, & |r| \leq \delta, \\ \delta\left(|r| - \frac{1}{2}\delta\right), & |r| > \delta. \end{cases} \tag{17}$$

The threshold $\delta$ controls the transition between quadratic and linear penalties, allowing small residuals to be strongly suppressed while preventing large but noisy deviations from dominating the loss. In practice, $\delta$ is set proportional to the empirical standard deviation of non-DEG effects, and is fixed across perturbations.

**Adaptive subgraph representation alignment.** Beyond expression-level supervision, we explicitly guide the learned subgraph representation to reflect perturbation-specific responses. Let $\mathbf{z}_{\text{context}}^{(p)}$ denote the context representation produced by the extracted subgraph for perturbation $p$ (Section 3.3). We construct a response-driven target by summa-

rizing DEG signals:

$$\mathbf{y}^{(p)} \in \mathbb{R}^N, \qquad \mathbf{y}_i^{(p)} = \begin{cases} \Delta\mathbf{X}_i^p, & i \in \mathcal{D}(p), \\ 0, & i \in \bar{\mathcal{D}}(p), \end{cases} \qquad (18)$$

which preserves signed effect sizes while masking non-DEG genes. This vector is mapped into the representation space via a projection head $g(\cdot)$:

$$\mathbf{t}^{(p)} = g\Big(\mathbf{y}^{(p)}\Big) \in \mathbb{R}^d. \qquad (19)$$

We then align the subgraph context with the response-driven target using a cosine-distance loss:

$$\mathcal{L}_{\text{align}} = \mathbb{E}\left[\left\|\frac{\mathbf{z}_{\text{context}}^{(p)}}{\|\mathbf{z}_{\text{context}}^{(p)}\|_2} - \frac{\mathbf{t}^{(p)}}{\|\mathbf{t}^{(p)}\|_2}\right\|_2^2\right]. \qquad (20)$$

This alignment encourages the extracted subgraph to encode perturbation-relevant DEG structure, rather than generic graph features.

**Overall objective.** The final training objective combines all three terms:

$$\mathcal{L}_{\text{total}} = \mathcal{L}_{\text{recon}} + \lambda_{\text{non}}\mathcal{L}_{\text{non}} + \lambda_{\text{align}}\mathcal{L}_{\text{align}}. \qquad (21)$$

# 4. Experiments

## 4.1. Experiment Setup

We evaluate ADAPERT on a single-cell genetic perturbation prediction task. The goal is to predict the transcriptional response of cells after a target gene is perturbed. All experiments are conducted under the *unseen perturbation* setting, where perturbations in the test set are not observed during training. We evaluate ADAPERT on four single-cell CRISPR perturbation datasets from Replogle *et al.* (Replogle et al., 2022), spanning the K562, RPE1, HEPG2, and JURKAT cell lines. For clarity, the main text reports results on two representative cell lines: **K562.Replogle** and **RPE1.Replogle**, each consisting of single-gene knockouts measured by single-cell RNA sequencing. Both datasets include control cells and perturbed cells for each target gene, enabling direct evaluation of perturbation-induced transcriptional changes. Complete results on all four datasets, together with a cross-cell-line generalization study, are reported in Section C. For each dataset, we follow standard preprocessing and data splitting protocols used in prior work. Training, validation, and test sets are constructed such that perturbations in the test set are entirely unseen during training. More details about the datasets are provided in the Appendix.

## 4.2. Baselines and Training Protocol

We compare ADAPERT against two categories of baseline methods: (1) models without a knowledge graph, including scVI (Lopez et al., 2018), CPA (Lotfollahi et al., 2023), and STATE (Adduri et al., 2025); and (2) models that incorporate a knowledge graph, including GEARS (Roohani et al., 2024), TxPert (Wenkel et al., 2025), and MorPH (He et al., 2025). These baselines represent state-of-the-art approaches for genetic perturbation prediction. All models are trained under the same experimental setup and computational budget to ensure fair comparison. Unless otherwise specified, we use identical data splits, training procedures, and evaluation protocols across all methods. Additional implementation details are provided in Appendix.

## 4.3. Evaluation Metrics

To evaluate perturbation prediction performance, we use a set of complementary metrics that capture different aspects of model behavior. We report two global metrics, Pearson-$\Delta$ and the Perturbation Discrimination Score (PDS), which measure overall agreement between predicted and observed perturbation effects. In addition, we include DEG-aware metrics that focus on differential expression accuracy, including Differential Expression Score@K (DES@K) and Spearman correlation of log fold changes and their directions. These metrics are sensitive to false positive predictions and better reflect the recovery of perturbation-specific gene responses. Together, this metric suite allows us to assess both global reconstruction accuracy and the ability to capture biologically meaningful perturbation effects. All metrics are computed using the latest version of the `cell-eval` (Adduri et al., 2025) evaluation framework.

# 5. Main Results

## 5.1. Global Performance of Perturbation Prediction

We conduct genetic perturbation prediction on two CRISPR perturbation datasets, **K562.Replogle** and **RPE1.Replogle**. We report Pearson correlation on differential expression relative to control (Pearson-$\Delta$) and the perturbation discriminative score (PDS). PDS measures how well a model distinguishes different perturbations. As shown in Table 1, methods without biological knowledge graphs show limited performance on both datasets. Their PDS values are close to chance level, indicating weak ability to separate different perturbations. Methods that use biological knowledge graphs perform better, showing that the use of biological prior knowledge is important. However, these methods rely on dense and mostly static graph structures, which can still spread noise across genes. ADAPERT achieves the best performance on both datasets. The gains are consistent for Pearson-$\Delta$ and PDS. Importantly, the improvement on

*Table 1.* Performance comparison across perturbation datasets. Results are reported as mean $\pm$ std over all test perturbations. $\Delta$ denotes correlation on differential expression relative to control, PDS denotes perturbation discriminative score.

| Category | Method | K562.Replogle | | RPE1.Replogle | |
|---|---|---|---|---|---|
| | | Pearson-$\Delta$ | PDS | Pearson-$\Delta$ | PDS |
| w/o KG | scVI (Lopez et al., 2018) | $0.171 \pm 0.195$ | $0.502 \pm 0.290$ | $0.369 \pm 0.175$ | $0.501 \pm 0.289$ |
| | CPA (Lotfollahi et al., 2023) | $0.288 \pm 0.157$ | $0.503 \pm 0.290$ | $0.486 \pm 0.210$ | $0.503 \pm 0.290$ |
| | STATE (Adduri et al., 2025) | $0.247 \pm 0.188$ | $0.506 \pm 0.289$ | $0.583 \pm 0.281$ | $0.520 \pm 0.292$ |
| w/ KG | GEARS (Roohani et al., 2024) | $0.298 \pm 0.181$ | $0.529 \pm 0.287$ | $0.396 \pm 0.216$ | $0.524 \pm 0.298$ |
| | TxPert (Wenkel et al., 2025) | $0.580 \pm 0.255$ | $0.665 \pm 0.310$ | $0.655 \pm 0.300$ | $0.618 \pm 0.290$ |
| | MorPH (He et al., 2025) | $0.442 \pm 0.242$ | $0.664 \pm 0.299$ | $0.541 \pm 0.299$ | $\mathbf{0.688 \pm 0.268}$ |
| | ADAPERT | $\mathbf{0.619 \pm 0.262}$ | $\mathbf{0.711 \pm 0.296}$ | $\mathbf{0.674 \pm 0.291}$ | $0.663 \pm 0.281$ |

*Table 2.* DEG-aware evaluation on **K562.Replogle**. Results are reported as mean $\pm$ std over multiple runs. DEG overlap@$k$ measures the fraction of ground-truth DEGs recovered in the top-$k$ predicted genes.

| Category | Method | Differential Expressed Score | | DE-metrics | | |
|---|---|---|---|---|---|---|
| | | @50 | @100 | Spearman-sig | Spearman-lfc-sig | Direction-match |
| w/o KG | scVI | $0.071 \pm 0.072$ | $0.064 \pm 0.060$ | 0.443 | $0.372 \pm 0.355$ | $0.577 \pm 0.186$ |
| | CPA | $0.116 \pm 0.098$ | $0.104 \pm 0.083$ | 0.474 | $0.360 \pm 0.104$ | $0.645 \pm 0.104$ |
| | STATE | $0.044 \pm 0.041$ | $0.050 \pm 0.045$ | 0.468 | $0.298 \pm 0.198$ | $0.643 \pm 0.108$ |
| w/ KG | GEARS | $0.126 \pm 0.107$ | $0.124 \pm 0.106$ | 0.540 | $0.381 \pm 0.341$ | $0.603 \pm 0.177$ |
| | TxPert | $0.220 \pm 0.173$ | $0.205 \pm 0.165$ | 0.536 | $0.640 \pm 0.241$ | $0.843 \pm 0.133$ |
| | MorPH | $0.057 \pm 0.083$ | $0.090 \pm 0.097$ | 0.546 | $0.448 \pm 0.239$ | $0.757 \pm 0.128$ |
| | ADAPERT | $\mathbf{0.263 \pm 0.190}$ | $\mathbf{0.252 \pm 0.182}$ | $\mathbf{0.622}$ | $\mathbf{0.688 \pm 0.240}$ | $\mathbf{0.867 \pm 0.141}$ |

PDS is larger than that on Pearson-$\Delta$. This shows that ADAPERT improves perturbation discrimination, rather than only increasing global correlation. The results suggest that ADAPERT reduces mean-collapsed predictions and focuses on perturbation-specific transcriptional changes.

### 5.2. DEG-aware Comparisons

As shown in Table 2, we report the Differential Expression Score (DES@K), which measures how well true DEGs are ranked among top predicted genes. Methods without knowledge graphs achieve very low DES, indicating poor separation between signal and noise. KG-based methods improve DEG recovery. GEARS shows moderate gains, and TxPert further improves DES, but its scores remain limited, suggesting that noise still affects non-DEG genes. ADAPERT achieves the highest DES at both $k = 50$ and $k = 100$, with consistent improvements over TxPert. This indicates more accurate ranking of true DEGs. ADAPERT also performs better on DEG-specific metrics, including Spearman correlation, the agreement of log-fold changes, and direction consistency. These results show that ADAPERT produces sparse and reliable perturbation-specific gene responses.

### 5.3. Comparison on Mean-Collapse

We analyze model sensitivity to mean-collapse by grouping perturbations into small-, medium-, and large-effect sets based on ground-truth effect size. Results are shown in Table 3. For small-effect perturbations, where mean-collapse is most severe, TxPert and ADAPERT achieve similar Pearson-$\Delta$, but ADAPERT shows much higher DES. This shows better separation of true signal from noise, despite similar overall correlation. For medium- and large- effect perturbations, ADAPERT improves all metrics, including Pearson-$\Delta$, DES, and PDS. Overall, these results show that ADAPERT is more robust to mean-collapse, especially when true perturbation effects are weak.

*Table 3.* Sensitivity analysis to mean bias under different perturbation effect sizes on **K562.Replogle** (mean $\pm$ std). Effect sizes are stratified by ground-truth differential expression. Improvements are reported relative to TxPert.

| Effect size | Model | P-$\Delta$ | DES | PDS |
|---|---|---|---|---|
| **Small-effect** ($<5\%$) | TxPert | 0.457 | 0.090 | 0.740 |
| | ADAPERT | 0.462 | 0.115 | 0.737 |
| | *Improvement* | $\uparrow 1.1\%$ | $\uparrow 28\%$ | $\downarrow 0.4\%$ |
| **Medium-effect** ($5\%$–$10\%$) | TxPert | 0.541 | 0.173 | 0.663 |
| | ADAPERT | 0.623 | 0.222 | 0.740 |
| | *Improvement* | $\uparrow 15\%$ | $\uparrow 28\%$ | $\uparrow 12\%$ |
| **Large-effect** ($>10\%$) | TxPert | 0.741 | 0.353 | 0.592 |
| | ADAPERT | 0.771 | 0.420 | 0.657 |
| | *Improvement* | $\uparrow 4.0\%$ | $\uparrow 19\%$ | $\uparrow 11\%$ |

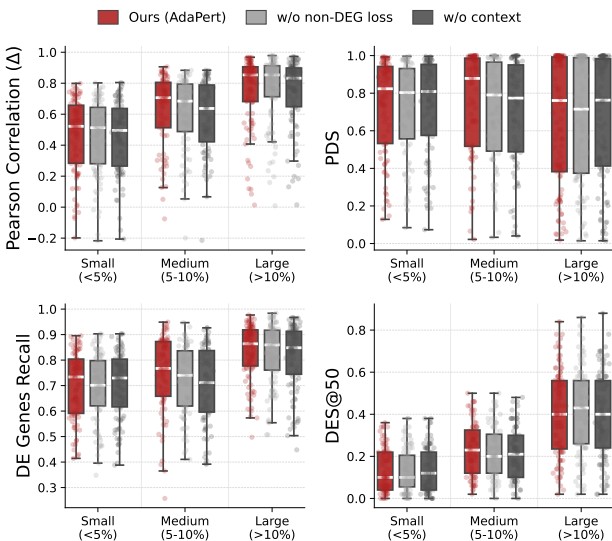

*Figure 4.* Comparison of model variants across small, medium, and large perturbations using global and DEG-based metrics.

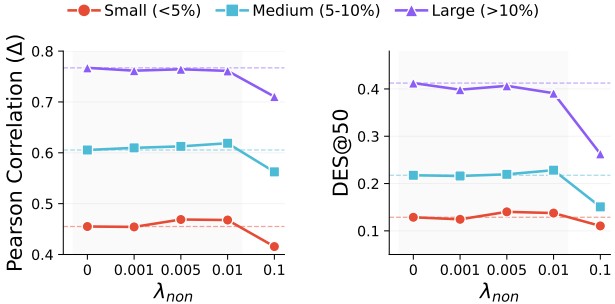

*Figure 5.* Effect-Size–Dependent Behavior of $\mathcal{L}_{\text{non}}$. Sensitivity of model performance to the weight $\lambda_{non}$ across small, medium, and large perturbations, evaluated using pearson$\Delta$ and DES@50.

## 5.4. Effect of $\mathcal{L}_{\text{non}}$ and $\mathbf{z}_{\text{context}}$ Across Perturbations

We conduct an ablation study on three perturbation groups categorized by effect size to evaluate the roles of perturbation-specific context $\mathbf{z}_{\text{context}}$ and the non-DEG loss $\mathcal{L}_{\text{non}}$ (Figure 4). Overall, the full model performs well across all metrics and effect-size regimes, with the strongest performance observed for small and medium perturbations. Removing $\mathbf{z}_{\text{context}}$ consistently degrades performance across all comparisons, showing that enriching perturbation context is critical. Removing the $\mathcal{L}_{\text{non}}$ has a strong negative impact for small and medium perturbations, where signals are weak and noise is high, indicating that adaptive separation of signal and noise is necessary in this regime. For large perturbations ($> 10\%$ DE genes), the effect of the $\mathcal{L}_{\text{non}}$ becomes less pronounced, showing that the balance between signal and noise varies with perturbation effect size.

## 5.5. Effect-Size–Dependent Behavior of the $\mathcal{L}_{\text{non}}$

We analyze the interaction between $\mathcal{L}_{\text{non}}$ and perturbation effect size by varying the weight $\lambda_{non}$ of the non-DEG loss across different perturbation groups (Figure 5). For small and medium perturbations, performance consistently improves as $\lambda_{non}$ increases from $0 \rightarrow 0.01$ across both global and DEG-based metrics. This trend indicates that assigning more weight to the non-DEG loss helps suppress noise and improves signal recovery when perturbation effects are weak. In contrast, for large perturbations, smaller values of $\lambda_{non}$ yield better performance, suggesting that strong signals require less regularization. Across all perturbation groups, setting $\lambda_{non}$ too large (e.g., $\lambda = 0.1$) leads to clear performance degradation. This behavior suggests that excessive smoothing over-suppresses perturbation signals and harms both gene-level recovery and global reconstruction.

## 6. Conclusion

We identify mean-collapse as a key failure mode in perturbation prediction and propose ADAPERT to address it through perturbation-specific context modeling and adaptive signal–noise separation. AdaPert consistently improves performance across benchmarks, especially for perturbations with small effects, and reveals the effect-size–dependent role of regularization. These results highlight the importance of adaptive modeling for robust perturbation prediction.

## Impact Statement

This work advances computational modeling of single-cell perturbation responses, which can reduce the cost and scale of wet-lab experiments and accelerate biological discovery and therapeutic development. By explicitly addressing mean-collapse and improving prediction for small-effect perturbations, AdaPert may help prioritize experimental candidates more reliably. As with any predictive model in biology, computational predictions should be validated experimentally before informing downstream decisions, and we caution against over-reliance on model outputs in safety-critical or clinical settings. We do not foresee direct negative societal consequences specific to this work beyond those broadly associated with advances in machine learning.

## Acknowledgments

This work was supported by the InnoCORE program of the Ministry of Science and ICT (MSIT) (N10250153); by Institute of Information & Communications Technology Planning & Evaluation (IITP) grants funded by the Korea government (MSIT) (RS-2025-02304967, AI Star Fellowship (KAIST); RS-2019-II190075, Artificial Intelligence Graduate School Program (KAIST)); by National Research Foundation of Korea (NRF) grants funded by MSIT (RS-2022-NR072184; RS-2024-00436165, GRDC Cooperative Hub Program); and by the Korea Health Industry Development Institute (KHIDI) grant funded by the Ministry of Health & Welfare (MOHW) (N0425208).

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

## Appendix Contents

## A. Data Statistics

### A.1. Single-Cell Genetic Perturbation Data

Predicting how cells respond to genetic perturbations is a key problem in functional genomics (Shalem et al., 2015), supporting downstream tasks such as understanding gene function, dissecting regulatory effects, and identifying therapeutic targets. Recent single-cell screens, including Perturb-seq and CRISPR-based assays (Bock et al., 2022; Dixit et al., 2016; Replogle et al., 2022; Lotfollahi et al., 2023), measure gene expression across thousands of genes under many perturbation conditions, motivating computational models that predict transcriptional responses to perturbations that have not been experimentally tested (Bunne et al., 2024; Chevalley et al., 2022).

We evaluate on four publicly available genome-wide CRISPRi Perturb-seq datasets spanning the human cell lines *K562*, *RPE1*, *JURKAT*, and *HEPG2*; K562 and RPE1 are from Replogle et al. (2022), while JURKAT and HEPG2 are added to broaden the cellular context. For each perturbation we use differential expression relative to matched non-targeting controls as the prediction target, train on highly variable genes (HVGs) only, and split at the perturbation level so that held-out perturbations are unseen during training. Per-split statistics are summarized in Table 4. Within each dataset we further stratify perturbations into small-, medium-, and large-effect categories by tertiles of the number of differentially expressed genes (DEGs), where DEGs are identified at BH-adjusted Wilcoxon $p < 0.05$ versus matched controls; the

per-split, per-effect-size counts are reported in Table 5. Figures 6a and 6b illustrate this heterogeneity on K562 and RPE1: overall log1p expression is right-skewed in both (mean = 0.54 / 0.59) with DEGs biased toward higher expression—modestly in K562 (n = 3,015, mean = 0.64) and markedly in RPE1 (n = 228, mean = 1.34); the average DEG count per perturbation also grows steeply with effect size, from 65 / 5,000 HVGs (1.3%) for small-effect to 482 (9.6%) for large-effect in K562, and from 106 / 3,352 (3.1%) to 649 (19.4%) in RPE1, indicating that RPE1 is more responsive to genetic perturbation overall.

*Table 4.* Statistics of the four single-cell genome-wide CRISPRi Perturb-seq datasets used in this work. K562 and RPE1 are from Replogle et al. (2022); JURKAT and HEPG2 are additionally included to broaden the evaluation across diverse cellular contexts. For each dataset we report the number of cells, the number of single-gene perturbations, and the number of highly variable genes (HVGs) used as model inputs. Train/Val/Test splits are defined at the perturbation level so that cells from held-out perturbations are never seen during training. For K562, the shared non-targeting control cells are assigned to *Train* and counted as a single entry in # Perturbations; for the remaining datasets, control cells are stored separately and are not enumerated in any split.

| Dataset | Split | # Cells | # Perturbations | # HVGs |
|---|---|---|---|---|
| *K562* | Train | 111,770 | 734 | |
| | Val | 10,918 | 82 | 5,000 |
| | Test | 38,475 | 272 | |
| *RPE1* | Train | 74,474 | 771 | |
| | Val | 26,073 | 308 | 3,352 |
| | Test | 50,593 | 464 | |
| *JURKAT* | Train | 73,904 | 745 | |
| | Val | 29,571 | 298 | 3,352 |
| | Test | 41,393 | 448 | |
| *HEPG2* | Train | 46,816 | 826 | |
| | Val | 18,524 | 330 | 3,352 |
| | Test | 29,688 | 497 | |

*Table 5.* Effect-size–stratified statistics across the four cell lines. Within each cell line, perturbations are grouped into Small/Medium/Large by tertiles of the number of differentially expressed genes (DEGs), where DEGs are identified at a BH-adjusted Wilcoxon $p < 0.05$ relative to matched non-targeting control cells. Per-dataset tertiles are used because the cell lines have different HVG counts (5,000 for K562 vs. 3,352 for the others) and different baseline responsiveness to perturbation, so a single absolute DEG-count threshold would not be comparable across datasets. # Perts and # Cells per row are the number of perturbations and the total number of perturbed cells falling into each (cell line, split, effect-size) bucket; row totals over the three effect-size bins recover the per-split counts in Table 4. This stratification reflects the substantial heterogeneity in perturbation-response magnitude within each dataset and enables a fine-grained evaluation under both sparse and strong transcriptional effect regimes.

| Dataset | Effect Size | Train | | Val | | Test | |
|---|---|---|---|---|---|---|---|
| | | # Perts | # Cells | # Perts | # Cells | # Perts | # Cells |
| *K562* | Small | 239 | 25,125 | 27 | 2,841 | 96 | 9,809 |
| | Medium | 240 | 24,222 | 26 | 2,807 | 99 | 11,790 |
| | Large | 254 | 51,732 | 29 | 5,270 | 77 | 16,876 |
| *RPE1* | Small | 248 | 30,695 | 114 | 11,666 | 150 | 16,923 |
| | Medium | 262 | 18,035 | 100 | 6,569 | 154 | 16,407 |
| | Large | 261 | 25,744 | 94 | 7,838 | 160 | 17,263 |
| *JURKAT* | Small | 229 | 14,996 | 100 | 6,127 | 160 | 10,292 |
| | Medium | 266 | 24,259 | 93 | 7,736 | 146 | 13,351 |
| | Large | 250 | 34,649 | 105 | 15,708 | 142 | 17,750 |
| *HEPG2* | Small | 252 | 9,439 | 124 | 5,033 | 167 | 6,147 |
| | Medium | 288 | 14,294 | 103 | 5,502 | 168 | 10,327 |
| | Large | 286 | 23,083 | 103 | 7,989 | 162 | 13,214 |

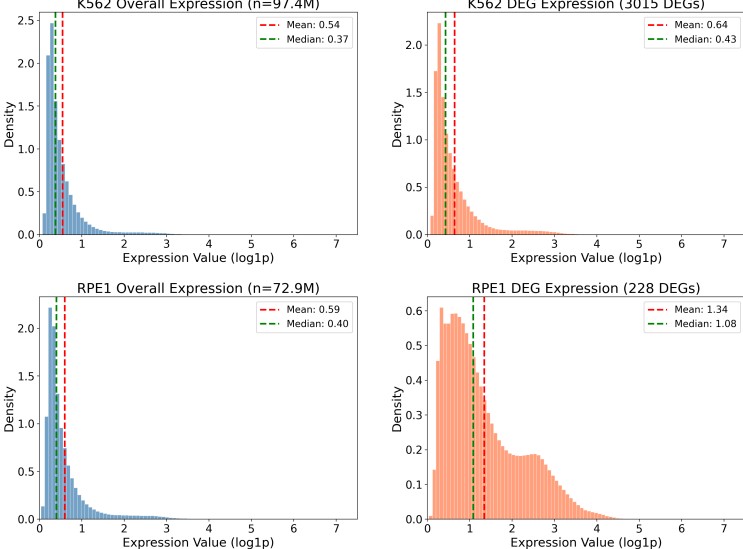

*(a)* **Expression distributions in K562 and RPE1.** (A-B) K562 cells show right-skewed overall expression (mean=0.54, median=0.37); the 3,015 DEGs have a modestly higher mean expression (mean=0.64, median=0.43). (C-D) RPE1 shows a similar overall pattern (mean=0.59, median=0.40), while its 228 DEGs are markedly higher and more symmetric (mean=1.34, median=1.08). Expression is log1p-transformed; RPE1 DEGs are computed from 50 perturbations using the top-20 genes ranked by absolute expression change.

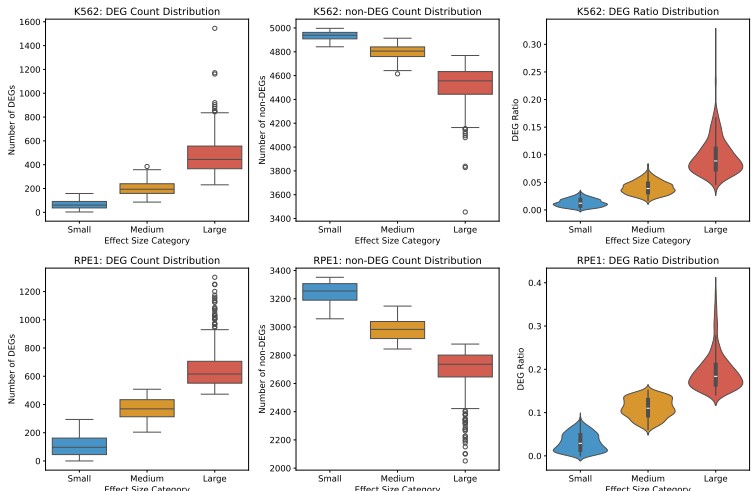

*(b)* **DEG distribution across perturbation effect-size categories.** (Top) Stacked bar plots of the mean number of DEGs (red) and non-DEGs (blue) per effect-size category in K562 (left) and RPE1 (right). (Bottom) Histograms of DEG counts across individual perturbations, colored by category (Small: blue, Medium: orange, Large: red). Categories are tertiles of mean absolute differential expression; DEGs are identified at an absolute expression-change threshold of 0.1.

## A.2. Knowledge Graph

We use a protein–protein interaction (PPI) knowledge graph constructed from STRING v11.5 (Szklarczyk et al., 2025; 2023). Nodes correspond to genes, and edges represent reported interactions between gene entities. The raw graph includes all interactions provided by STRING and is highly dense. To align the graph with the perturbation datasets, we restrict the graph to genes measured in the experiments. Specifically, we retain only highly variable genes (HVGs) used as model

inputs. This step substantially reduces graph size while preserving genes relevant to perturbation modeling. After HVG filtering, the graph remains dense. To further control graph complexity, we apply top-$k$ edge filtering, retaining only the $k$ highest-confidence edges per gene. We consider $k = 10$ and $k = 20$. Graph statistics for the raw graph, the HVG-filtered graph, and the top-$k$ graphs are reported in Table 6. Top-$k$ filtering substantially reduces node degree, yielding much sparser graph structures. This motivates learning perturbation-conditioned subgraphs from localized graph neighborhoods, rather than operating on the full dense graph.

*Table 6.* Statistics of the STRING knowledge graph. The raw graph contains all protein–protein interactions from STRING v11.5. HVG-filtered graphs are restricted to highly variable genes used in experiments. Top-$k$ variants retain the $k$ highest-confidence edges per gene.

| Graph | # Nodes | # Edges | Avg. Degree | Med. Degree |
|---|---|---|---|---|
| Raw (Full STRING) | 18,382 | 11,257,696 | 1,224.9 | 970 |
| HVG | 4,509 | 1,090,554 | 483.7 | 386 |
| HVG + Top-20 | 4,509 | 89,793 | 39.8 | 36 |
| HVG + Top-10 | 4,509 | 45,013 | 20.0 | 18 |

**DEG coverage in the knowledge graph.** We assess whether the knowledge graph captures perturbation-relevant genes by measuring DEG coverage in graph proximity to the perturbed gene. For each perturbation in the test set, we compute the fraction of true DEGs reachable within a small number of hops (e.g., 1–3) from the perturbed gene node. As shown in Figure 7, a large fraction of DEGs lie close to the perturbed gene in the graph. This supports the use of local graph context for perturbation modeling.

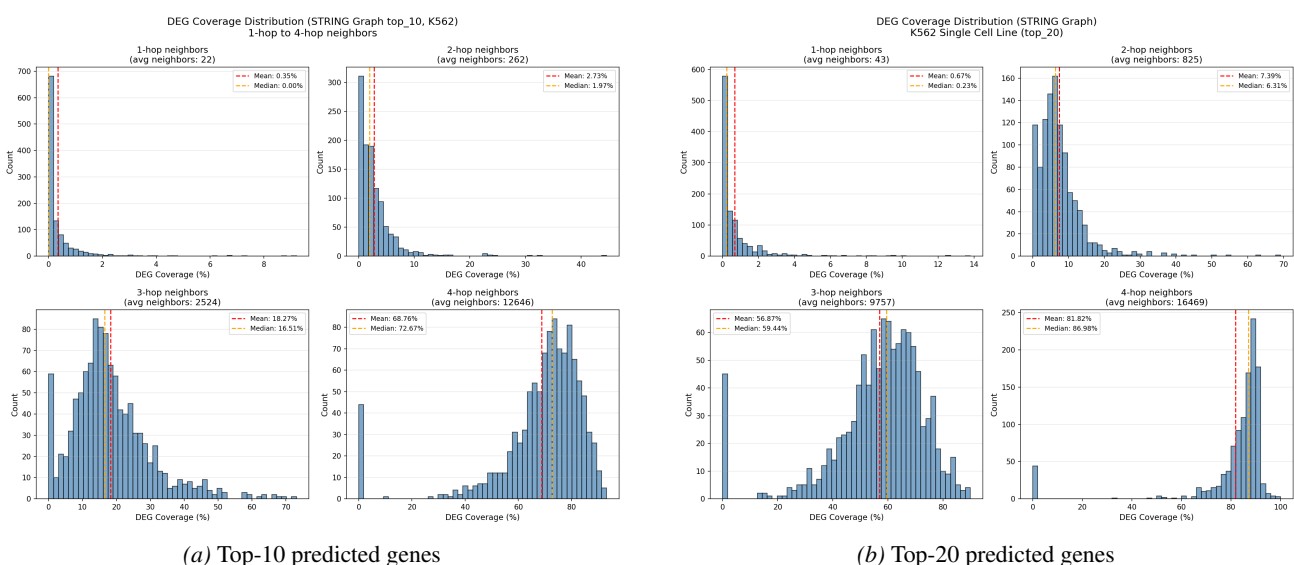

*(a)* Top-10 predicted genes        *(b)* Top-20 predicted genes

*Figure 7.* DEG coverage as a function of graph hop distance for different prediction depths. (a) Top-10 predicted genes. (b) Top-20 predicted genes.

### A.3. Gene Descriptions

We use GenePT (Chen & Zou, 2024) embeddings derived from NCBI and UniProt gene descriptions encoded via OpenAI's text embedding models (Dunefsky et al., 2024). The embeddings are available in two variants: Ada (1,536-dim) and Model 3 (3,072-dim), covering 93,800 and 133,736 genes respectively. Coverage for our datasets is high: 95.8% for K562 and 98.6% for RPE1 HVGs.

*Table 7.* Statistics and HVG coverage of GenePT-based gene embeddings.

| Statistic | Ada Embedding | Model 3 Embedding |
|---|---|---|
| Total genes | 93,800 | 133,736 |
| Embedding dimension | 1,536 | 3,072 |
| K562 HVG coverage | 4,790 / 5,000 (95.8%) | 4,790 / 5,000 (95.8%) |
| RPE1 HVG coverage | 3,304 / 3,352 (98.6%) | 3,304 / 3,352 (98.6%) |

## B. Experiment Settings

### B.1. Baseline Details

We compare ADAPERT with a set of representative baselines for single-cell genetic perturbation modeling. These baselines differ in model design, conditioning strategy, and the use of biological prior knowledge.

#### B.1.1. BASELINES WITHOUT BIOLOGICAL KNOWLEDGE GRAPHS

**scVI** (Lopez et al., 2018) is a variational autoencoder for single-cell RNA-seq data that learns a latent representation of gene expression without explicit conditioning on perturbations. It models the distribution of expression counts using a probabilistic decoder and serves as a purely data-driven baseline for comparing latent generative approaches.

**CPA** (Lotfollahi et al., 2023) (Compositional Perturbation Autoencoder) learns disentangled latent representations of control and perturbed cells. It separates a cell's basal state from perturbation effects in the latent space, enabling prediction of unseen perturbations and combinations. CPA can also learn interpretable embeddings for cells and perturbations and supports out-of-distribution predictions by recombining learned latent factors.

**STATE** (Adduri et al., 2025) is a deep generative model designed to predict perturbation effects on single-cell expression by transforming latent representations in a structured space. It accounts for cellular heterogeneity and aims to capture complex, nonlinear responses across conditions.

#### B.1.2. BASELINES WITH BIOLOGICAL KNOWLEDGE GRAPHS

**GEARS** (Roohani et al., 2024) integrates protein–protein interaction information into perturbation prediction by using graph-based message passing to propagate perturbation signals over network structure. This allows the model to leverage known gene interaction topology when predicting expression changes.

**TxPert** (Wenkel et al., 2025) uses graph representations of biological relationships to inform prediction of transcriptional responses under out-of-distribution settings. It conditions expression prediction on graph-based embeddings that capture biochemical relationships among genes, enabling generalization to unseen perturbations and cell contexts.

**MORPH** (He et al., 2025) combines a discrepancy-based variational autoencoder with an attention mechanism to predict cellular responses to unseen perturbations, including unseen single genes, perturbation combinations, and cell contexts. The attention mechanism enables the model to infer gene interactions and regulatory effects while learning latent perturbation representations.

All baselines are evaluated using their recommended settings and official implementations when available. We apply the same data splits, preprocessing, and evaluation protocols across all methods to ensure fair comparison.

### B.2. Metric Definitions

Let $\mathbf{X}^c, \mathbf{X}^p \in \mathbb{R}^N$ denote the control and perturbed expression profiles, and let $\hat{\mathbf{X}}^p$ be the predicted perturbed profile. We define the true and predicted perturbation effects as

$$\Delta\mathbf{X}^p = \mathbf{X}^p - \mathbf{X}^c, \qquad \Delta\hat{\mathbf{X}}^p = \hat{\mathbf{X}}^p - \mathbf{X}^c. \tag{22}$$

### B.2.1. GLOBAL METRICS

**Pearson-$\Delta$.** We compute the Pearson correlation between the predicted and true perturbation effects:

$$\text{Pearson-}\Delta = \text{corr}\left(\Delta\hat{\mathbf{X}}^p, \Delta\mathbf{X}^p\right). \tag{23}$$

This metric measures global agreement in perturbation-induced expression changes.

**Perturbation Discrimination Score (PDS).** To evaluate whether predicted perturbation effects are specific to the correct perturbation, we use the Perturbation Discrimination Score (PDS) following the Virtual Cell Challenge. For each perturbation $p$, we compute the distance between its predicted effect $\Delta\hat{\mathbf{X}}^p$ and the true effects of all perturbations in the test set:

$$d_{p,t} = \left\|\Delta\hat{\mathbf{X}}^p - \Delta\mathbf{X}^t\right\|_1, \qquad \forall t \in \mathcal{T}, \tag{24}$$

where $\mathcal{T}$ denotes the set of test perturbations.

We rank these distances in ascending order and define the rank of the correct perturbation as

$$r_p = 1 + \sum_{t \neq p} \mathbb{I}[d_{p,t} < d_{p,p}]. \tag{25}$$

The discrimination score for perturbation $p$ is then

$$\text{PDS}_p = 1 - \frac{r_p - 1}{|\mathcal{T}|}. \tag{26}$$

The final PDS is obtained by averaging $\text{PDS}_p$ over all perturbations in the test set. Higher values indicate better discrimination of perturbation-specific effects.

Let $\mathcal{D}(p)$ denote the set of differentially expressed genes for perturbation $p$, defined using the ground-truth data.

### B.2.2. DEG-AWARE METRICS.

**Differential Expression Score (DES).** Following the Virtual Cell Challenge evaluation, we assess whether a model recovers the correct set of differentially expressed genes after perturbation. For each perturbation $p$, let $G_{\text{true}}(p)$ denote the ground-truth set of significant DEGs and $G_{\text{pred}}(p)$ the predicted set of significant DEGs, both defined at a fixed false discovery rate threshold.

The Differential Expression Score for perturbation $p$ is defined as the fraction of true DEGs that are recovered in the predicted set:

$$\text{DES}(p) = \frac{|G_{\text{true}}(p) \cap G_{\text{pred}}(p)|}{|G_{\text{true}}(p)|}. \tag{27}$$

The overall DES is obtained by averaging $\text{DES}(p)$ over all perturbations in the test set. Higher values indicate better recovery of differentially expressed genes.

**DE-Spearman (significant genes).** We compute the Spearman rank correlation between predicted and true effects over DEGs:

$$\text{DE-Spearman-sig} = \rho_s(\Delta\hat{\mathbf{x}}_{\mathcal{D}}^p, \Delta\mathbf{x}_{\mathcal{D}}^p). \tag{28}$$

**DE-Spearman (LFC-weighted).** To emphasize genes with larger effect sizes, we compute a weighted Spearman correlation using absolute ground-truth effects as weights:

$$\text{DE-Spearman-lfc-sig} = \rho_s^{(w)}(\Delta\hat{\mathbf{x}}_{\mathcal{D}}^p, \Delta\mathbf{x}_{\mathcal{D}}^p, |\Delta\mathbf{x}_{\mathcal{D}}^p|). \tag{29}$$

**DE Direction Match.** We measure the fraction of DEGs for which the predicted and true effect directions agree:

$$\text{DE-Dir} = \frac{1}{|\mathcal{D}(p)|} \sum_{i \in \mathcal{D}(p)} \mathbb{I}\left[\text{sign}(\Delta\hat{\mathbf{X}}_i^p) = \text{sign}(\Delta\mathbf{X}_i^p)\right]. \tag{30}$$

## B.3. Model Inference Detail

We illustrate the full inference pipeline of ADAPERT for an unseen perturbation gene in Figure 8.

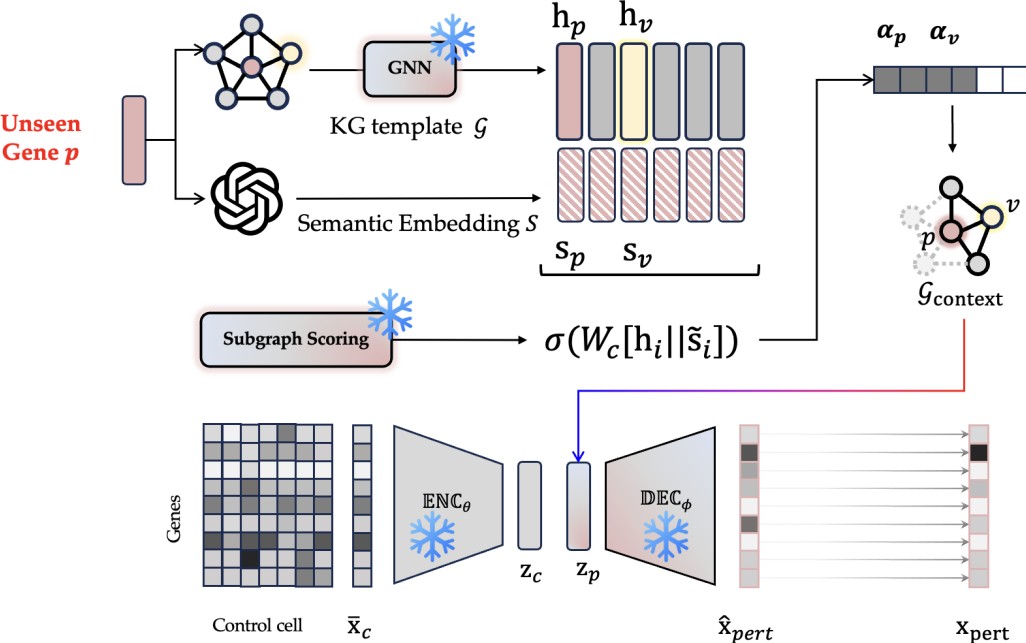

*Figure 8.* **Detailed illustration of ADAPERT's adaptive learning step at inference for an unseen perturbation gene. (Top-left)** The perturbation gene is located on a shared KG template (STRING PPI network). A 4-layer GATv2 with skip-cat connections produces structural node embeddings, while a pre-trained language model (GenePertESM) provides semantic embeddings. **(Middle)** A subgraph scoring function computes perturbation-conditioned attention weights for each neighbor by jointly leveraging structural and semantic features. Neighbors exceeding a threshold are selected to form the context subgraph, which adaptively varies in size per perturbation. **(Bottom)** The encoder maps control-cell expression to a basal-state representation. The context vector is computed via attention-weighted sum pooling over selected neighbors. The decoder takes the concatenation of basal state, context vector, and semantic embedding to predict perturbed gene expression. Crucially, the GNN and subgraph scoring operate on a fixed KG template shared across all perturbations— the adaptive step is purely attention-based, requiring no perturbation-specific graph construction at inference time.

## C. Comprehensive Analysis on Perturbation Prediction

We provide a comprehensive comparison of ADAPERT against all baselines on four single-cell perturbation datasets— K562, JURKAT, HEPG2, and RPE1— under the unseen-perturbation setting, and additionally evaluate cross-cell-line generalization by training on three cell lines and testing on the held-out K562 cell line. Each method is assessed with six metrics spanning global accuracy (Pearson-$\Delta$, MAE-$\Delta$, Discrimination L1) and DEG-level fidelity (Overlap@50, DE Direction Match, DE Spearman LFC). Figures 9–13 visualize per-metric comparisons and overall rankings, and Tables 9–13 report the corresponding numerical values.

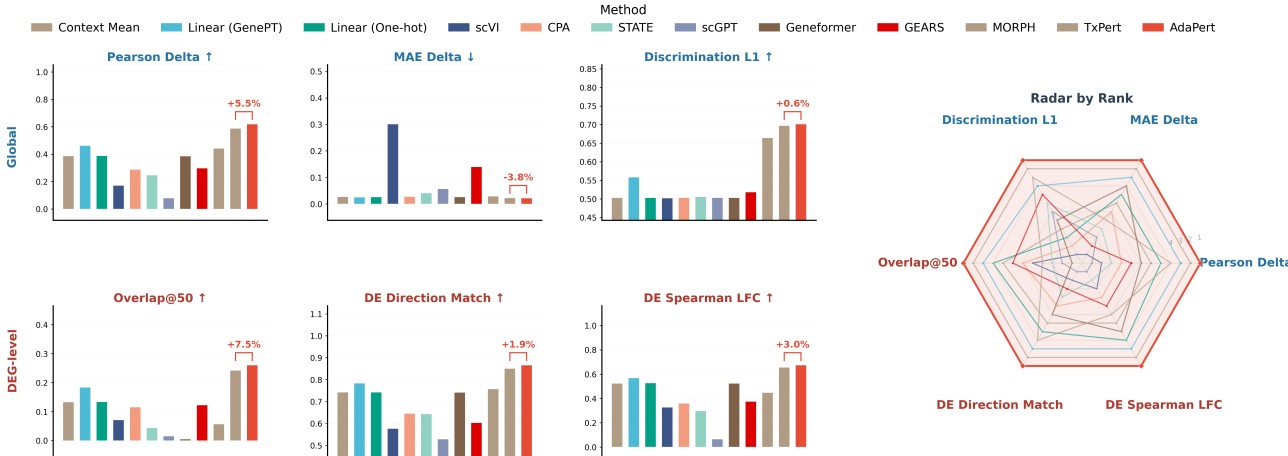

*Figure 9.* **Method comparison on the K562 unseen-perturbation setting** ($n = 272$ **test perturbations**). Top row: global metrics (Pearson-$\Delta$, MAE-$\Delta$, Discrimination L1). Bottom row: DEG-level metrics (Overlap@50, DE Direction Match, DE Spearman LFC). Right: radar chart ranking all methods across the six metrics. ADAPERT achieves the highest Pearson-$\Delta$ with a $+5.5\%$ improvement over the second-best method (TxPert), along with the best Overlap@50 ($+7.5\%$), MAE-$\Delta$ ($-3.8\%$), DE Spearman LFC ($+3.0\%$), and DE Direction Match ($+1.9\%$), ranking first across all evaluated metrics. Percentage annotations indicate ADAPERT's relative improvement over the next-best method.

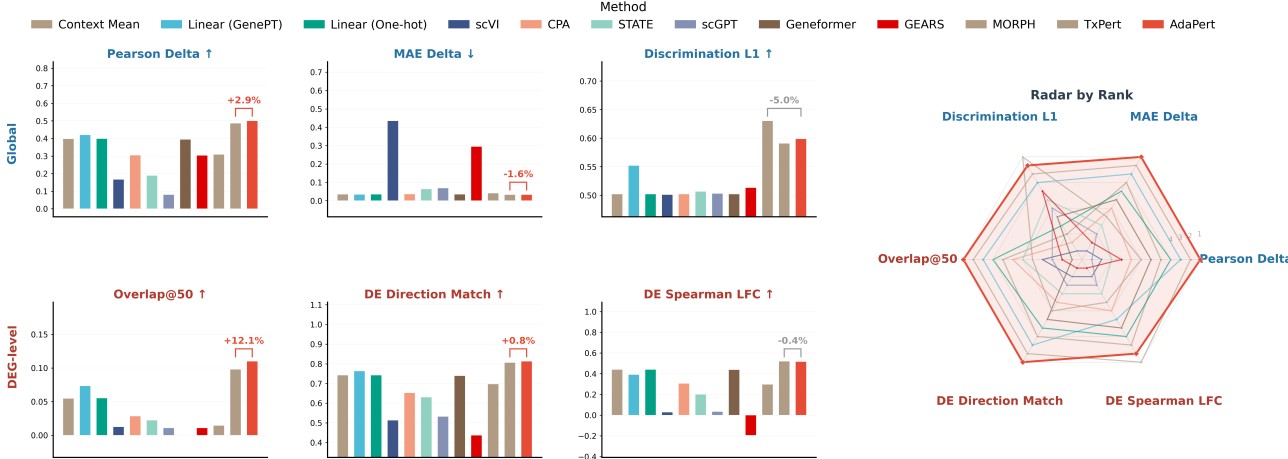

*Figure 10.* **Method comparison on the JURKAT unseen-perturbation setting** ($n = 446$ **test perturbations**). Top row: global metrics (Pearson-$\Delta$, MAE-$\Delta$, Discrimination L1). Bottom row: DEG-level metrics (Overlap@50, DE Direction Match, DE Spearman LFC). Right: radar chart ranking all methods across the six metrics. ADAPERT achieves the highest Pearson-$\Delta$ with a $+2.9\%$ improvement over TxPert, along with the best MAE-$\Delta$ ($-1.6\%$), Overlap@50 ($+12.1\%$), and DE Direction Match ($+0.8\%$). Percentage annotations indicate ADAPERT's relative improvement over the next-best method.

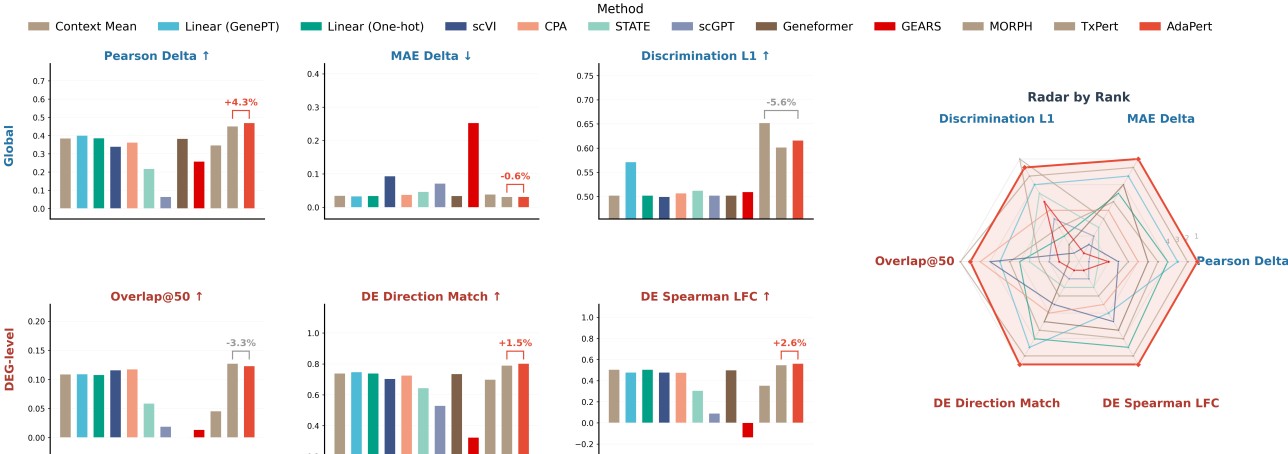

*Figure 11.* **Method comparison on the HEPG2 unseen-perturbation setting ($n = 496$ test perturbations).** Top row: global metrics (Pearson-$\Delta$, MAE-$\Delta$, Discrimination L1). Bottom row: DEG-level metrics (Overlap@50, DE Direction Match, DE Spearman LFC). Right: radar chart ranking all methods across the six metrics. ADAPERT achieves the highest Pearson-$\Delta$ with a $+4.3\%$ improvement over TxPert, along with the best MAE-$\Delta$ ($-0.6\%$), DE Direction Match ($+1.5\%$), and DE Spearman LFC ($+2.6\%$). Percentage annotations indicate ADAPERT's relative improvement over the next-best method.

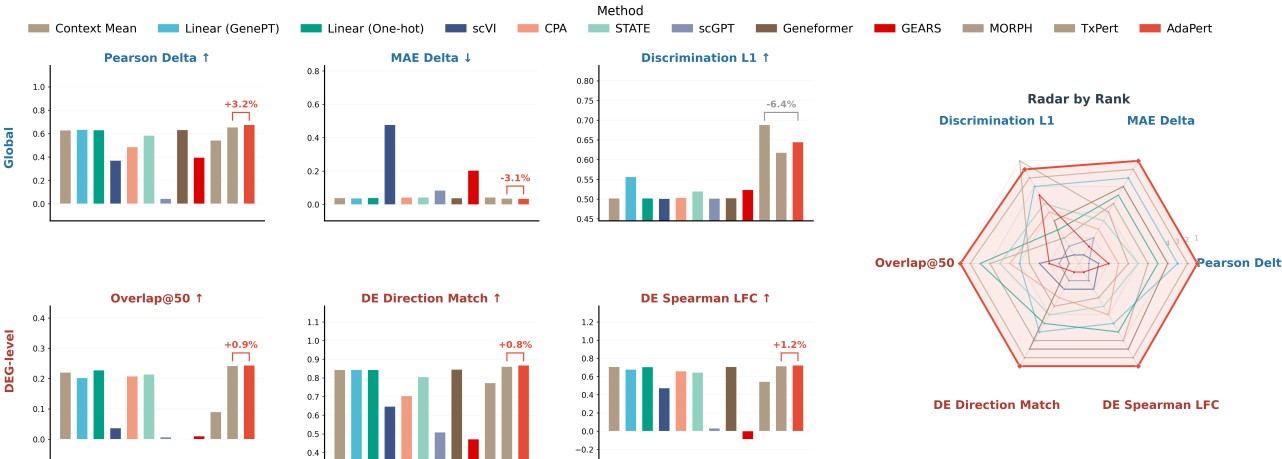

*Figure 12.* **Method comparison on the RPE1 unseen-perturbation setting ($n = 461$ test perturbations).** Top row: global metrics (Pearson-$\Delta$, MAE-$\Delta$, Discrimination L1). Bottom row: DEG-level metrics (Overlap@50, DE Direction Match, DE Spearman LFC). Right: radar chart ranking all methods across the six metrics. ADAPERT achieves the highest Pearson-$\Delta$ with a $+3.2\%$ improvement over TxPert, along with the best MAE-$\Delta$ ($-3.1\%$), Overlap@50 ($+0.9\%$), DE Direction Match ($+0.8\%$), and DE Spearman LFC ($+1.2\%$). Percentage annotations indicate ADAPERT's relative improvement over the next-best method.

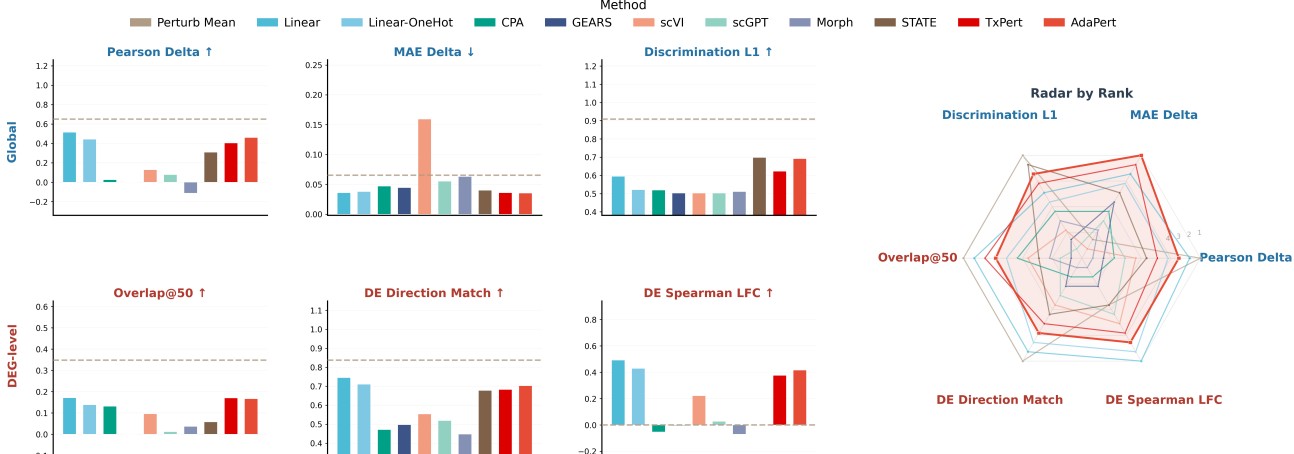

*Figure 13.* **Cross-cell-line evaluation on the K562 cell line.** Models are trained on three cell lines (RPE1, HEPG2, JURKAT) and evaluated on the held-out K562 cell line. Eleven methods are compared: Perturb Mean (dashed line, perturbation-aware training-set average), Linear-GenePT, Linear-OneHot, CPA, GEARS, scVI, scGPT, MORPH, STATE, TxPert, and ADAPERT. **(Top row)** Global metrics: Pearson-$\Delta$, MAE-$\Delta$, Discrimination L1. **(Bottom row)** DEG-level metrics: Overlap@50, DE Direction Match, DE Spearman LFC. **(Right)** Radar chart ranking all methods across the six metrics. Perturb Mean (dashed line) achieves the highest Pearson-$\Delta$, reflecting that the majority of perturbation effects are conserved across cell lines (59% of test perturbations have Perturb Mean $\Delta \geq 0.7$). Linear-GenePT (0.515) outperforms all other learning-based methods on Pearson-$\Delta$, likely benefiting from LLM-derived embeddings that encode cell-type-independent functional priors. ADAPERT achieves the best performance among deep-learning models (Pearson-$\Delta = 0.460$, MAE-$\Delta = 0.035$). We note that ADAPERT is designed for the unseen-perturbation setting; the cross-cell-line setting introduces additional challenges related to cell-state variation and imbalanced perturbation conservation across training cell lines.

*Table 9.* All metrics on the K562 unseen-perturbation test set ($n = 272$). Best value per column in **bold**.

| Method | Pearson-$\Delta$ | MAE-$\Delta$ | Discrim. L1 | Overlap@50 | DE Dir. Match | DE Spearman LFC |
|---|---|---|---|---|---|---|
| Context Mean | 0.3867 | 0.0266 | 0.5030 | 0.1333 | 0.7419 | 0.5241 |
| Linear (GenePT) | 0.4629 | 0.0252 | 0.5584 | 0.1840 | 0.7836 | 0.5695 |
| Linear (One-hot) | 0.3886 | 0.0264 | 0.5028 | 0.1337 | 0.7421 | 0.5266 |
| scVI | 0.1711 | 0.3012 | 0.5020 | 0.0714 | 0.5769 | 0.3271 |
| CPA | 0.2877 | 0.0276 | 0.5028 | 0.1155 | 0.6454 | 0.3603 |
| scGPT | 0.0792 | 0.0567 | 0.5032 | 0.0154 | 0.5283 | 0.0657 |
| STATE | 0.2468 | 0.0412 | 0.5057 | 0.0442 | 0.6431 | 0.2977 |
| Geneformer | 0.3863 | 0.0263 | 0.5030 | 0.0050 | 0.7414 | 0.5247 |
| GEARS | 0.2975 | 0.1396 | 0.5182 | 0.1226 | 0.6034 | 0.3761 |
| MORPH | 0.4415 | 0.0292 | 0.6642 | 0.0569 | 0.7573 | 0.4479 |
| TxPert | 0.5879 | 0.0229 | 0.6971 | 0.2426 | 0.8502 | 0.6552 |
| ADAPERT | **0.6200** | **0.0220** | **0.7014** | **0.2607** | **0.8663** | **0.6745** |

*Table 10.* All metrics on the JURKAT unseen-perturbation test set ($n = 446$). Best value per column in **bold**.

| Method | Pearson-$\Delta$ | MAE-$\Delta$ | Discrim. L1 | Overlap@50 | DE Dir. Match | DE Spearman LFC |
|---|---|---|---|---|---|---|
| Context Mean | 0.3975 | 0.0348 | 0.5021 | 0.0547 | 0.7427 | 0.4409 |
| Linear (GenePT) | 0.4203 | 0.0343 | 0.5521 | 0.0735 | 0.7631 | 0.3924 |
| Linear (One-hot) | 0.3986 | 0.0349 | 0.5021 | 0.0552 | 0.7421 | 0.4405 |
| scVI | 0.1673 | 0.4351 | 0.5009 | 0.0124 | 0.5132 | 0.0284 |
| CPA | 0.3046 | 0.0362 | 0.5021 | 0.0285 | 0.6524 | 0.3057 |
| scGPT | 0.0797 | 0.0693 | 0.5031 | 0.0108 | 0.5322 | 0.0343 |
| STATE | 0.1894 | 0.0640 | 0.5066 | 0.0221 | 0.6304 | 0.2016 |
| Geneformer | 0.3947 | 0.0349 | 0.5022 | 0.0000 | 0.7400 | 0.4389 |
| GEARS | 0.3040 | 0.2948 | 0.5132 | 0.0107 | 0.4375 | −0.1935 |
| MORPH | 0.3091 | 0.0412 | **0.6303** | 0.0145 | 0.6975 | 0.2979 |
| TxPert | 0.4861 | 0.0332 | 0.5909 | 0.0981 | 0.8061 | **0.5198** |
| ADAPERT | **0.5003** | **0.0327** | 0.5990 | **0.1099** | **0.8127** | 0.5175 |

*Table 11.* All metrics on the HEPG2 unseen-perturbation test set ($n = 496$). Best value per column in **bold**.

| Method | Pearson-$\Delta$ | MAE-$\Delta$ | Discrim. L1 | Overlap@50 | DE Dir. Match | DE Spearman LFC |
|---|---|---|---|---|---|---|
| Context Mean | 0.3850 | 0.0347 | 0.5021 | 0.1088 | 0.7382 | 0.5043 |
| Linear (GenePT) | 0.4004 | 0.0332 | 0.5713 | 0.1090 | 0.7467 | 0.4773 |
| Linear (One-hot) | 0.3854 | 0.0345 | 0.5021 | 0.1079 | 0.7391 | 0.5052 |
| scVI | 0.3397 | 0.0933 | 0.4994 | 0.1157 | 0.7033 | 0.4778 |
| CPA | 0.3621 | 0.0376 | 0.5069 | 0.1173 | 0.7252 | 0.4753 |
| scGPT | 0.0640 | 0.0713 | 0.5022 | 0.0189 | 0.5294 | 0.0897 |
| STATE | 0.2180 | 0.0466 | 0.5121 | 0.0586 | 0.6435 | 0.3054 |
| Geneformer | 0.3822 | 0.0344 | 0.5019 | 0.0000 | 0.7349 | 0.4989 |
| GEARS | 0.2579 | 0.2529 | 0.5096 | 0.0132 | 0.3234 | −0.1385 |
| MORPH | 0.3464 | 0.0387 | **0.6522** | 0.0452 | 0.6983 | 0.3534 |
| TxPert | 0.4502 | 0.0316 | 0.6016 | **0.1272** | 0.7902 | 0.5473 |
| ADAPERT | **0.4694** | **0.0314** | 0.6158 | 0.1230 | **0.8022** | **0.5617** |

*Table 12.* All metrics on the RPE1 unseen-perturbation test set ($n = 461$). Best value per column in **bold**.

| Method | Pearson-$\Delta$ | MAE-$\Delta$ | Discrim. L1 | Overlap@50 | DE Dir. Match | DE Spearman LFC |
|---|---|---|---|---|---|---|
| Context Mean | 0.6288 | 0.0387 | 0.5019 | 0.2203 | 0.8434 | 0.7061 |
| Linear (GenePT) | 0.6334 | 0.0372 | 0.5564 | 0.2024 | 0.8433 | 0.6760 |
| Linear (One-hot) | 0.6302 | 0.0385 | 0.5021 | 0.2277 | 0.8431 | 0.7043 |
| scVI | 0.3693 | 0.4772 | 0.5010 | 0.0365 | 0.6472 | 0.4717 |
| CPA | 0.4861 | 0.0428 | 0.5034 | 0.2078 | 0.7029 | 0.6586 |
| scGPT | 0.0440 | 0.0841 | 0.5017 | 0.0070 | 0.5084 | 0.0315 |
| STATE | 0.5832 | 0.0427 | 0.5196 | 0.2139 | 0.8052 | 0.6431 |
| Geneformer | 0.6316 | 0.0383 | 0.5021 | 0.0000 | 0.8446 | 0.7064 |
| GEARS | 0.3963 | 0.2036 | 0.5235 | 0.0101 | 0.4715 | −0.0888 |
| MORPH | 0.5409 | 0.0424 | **0.6885** | 0.0901 | 0.7728 | 0.5437 |
| TxPert | 0.6545 | 0.0357 | 0.6177 | 0.2416 | 0.8602 | 0.7144 |
| ADAPERT | **0.6756** | **0.0346** | 0.6446 | **0.2438** | **0.8674** | **0.7230** |

*Table 13.* All metrics on the K562 cross-cell-line test set (trained on RPE1/HEPG2/JURKAT, tested on K562). Best value per column in **bold**. Perturb Mean is a perturbation-aware training-set average that serves as a strong upper bound when perturbation effects are conserved across cell lines.

| Method | Pearson-$\Delta$ | MAE-$\Delta$ | Discrim. L1 | Overlap@50 | DE Dir. Match | DE Spearman LFC |
|---|---|---|---|---|---|---|
| Perturb Mean | **0.6501** | 0.0654 | **0.9086** | **0.3480** | **0.8372** | **0.8170** |
| Linear-GenePT | 0.5149 | 0.0359 | 0.5950 | 0.1714 | 0.7449 | 0.4916 |
| Linear-OneHot | 0.4423 | 0.0378 | 0.5212 | 0.1386 | 0.7106 | 0.4289 |
| CPA | 0.0265 | 0.0471 | 0.5191 | 0.1315 | 0.4709 | −0.0525 |
| GEARS | −0.0021 | 0.0445 | 0.5024 | 0.0000 | 0.4972 | −0.0036 |
| scVI | 0.1291 | 0.1593 | 0.5025 | 0.0958 | 0.5536 | 0.2214 |
| scGPT | 0.0776 | 0.0550 | 0.5023 | 0.0117 | 0.5189 | 0.0279 |
| MORPH | −0.1126 | 0.0629 | 0.5106 | 0.0370 | 0.4468 | −0.0704 |
| STATE | 0.3088 | 0.0401 | 0.6984 | 0.0579 | 0.6776 | 0.0000 |
| TxPert | 0.4045 | 0.0358 | 0.6224 | 0.1702 | 0.6821 | 0.3763 |
| ADAPERT | 0.4601 | **0.0352** | 0.6916 | 0.1668 | 0.7027 | 0.4157 |

**Systema evaluation.** We additionally evaluate all methods under the Systema protocol, which scores predictions with both perturbation- and control-referenced Pearson-$\Delta$ (over all genes and the top-20 DE genes) and a centroid-based accuracy. Figures 14–18 and Tables 14–18 report these results across the four cell lines and the K562 cross-cell-line setting. ADAPERT obtains the best perturbation-referenced scores on every dataset, confirming the conclusions of the comprehensive analysis above under an independent evaluation protocol.

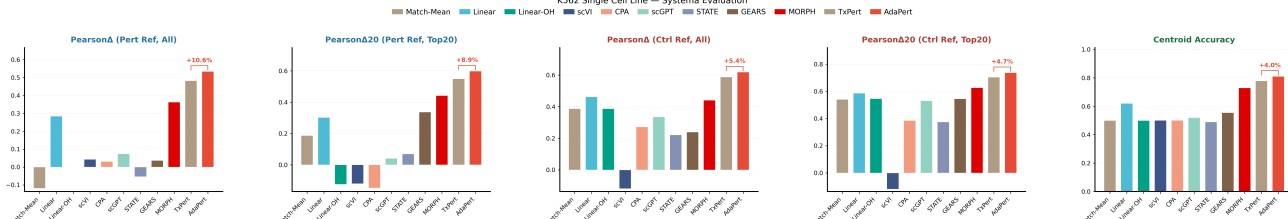

*Figure 14.* **Systema evaluation on the K562 unseen-perturbation setting ($n = 272$ perturbations, 12 methods).** Five metrics are shown: Pearson-$\Delta$ with perturbation reference (all genes and top-20 DE genes), Pearson-$\Delta$ with control reference (all genes and top-20), and centroid accuracy. ADAPERT achieves the best performance across all five metrics ($P\Delta_{pert} = 0.534$, $P\Delta20_{pert} = 0.599$, centroid $= 0.810$), substantially outperforming TxPert (0.483, 0.550, 0.779) and all other baselines. Notably, STATE ($-0.054$) and Matching-Mean ($-0.119$) produce negative $P\Delta_{pert}$, indicating predictions worse than random on this dataset.

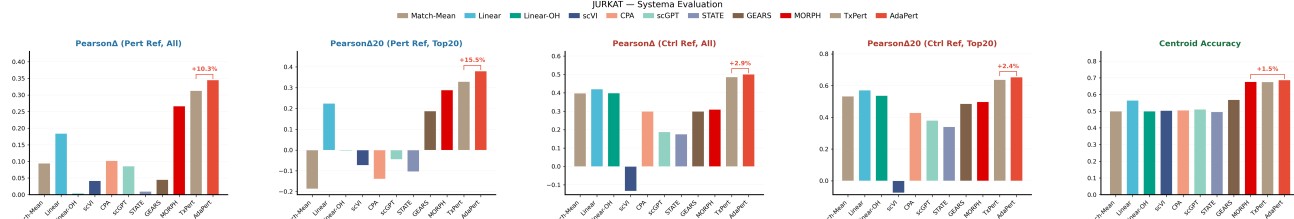

*Figure 15.* **Systema evaluation on the JURKAT unseen-perturbation setting ($n = 446$ perturbations).** ADAPERT achieves the best $P\Delta_{pert}$ (0.345), $P\Delta20_{pert}$ (0.380), and centroid accuracy (0.687). TxPert follows closely (0.313, 0.329, 0.675). MORPH is competitive on centroid accuracy (0.676) but lower on perturbation-reference metrics. Foundation models (scGPT, scVI) and STATE show limited effectiveness.

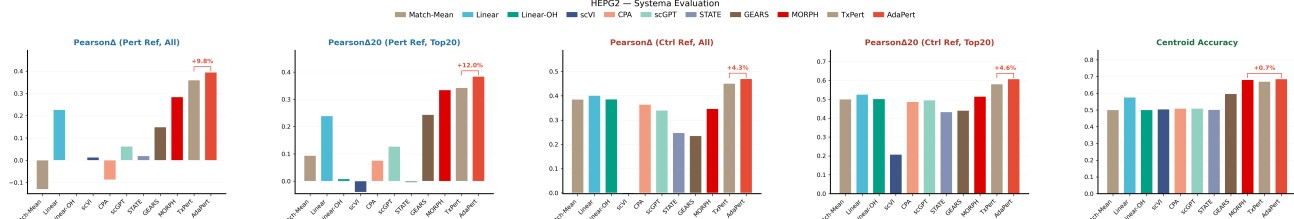

*Figure 16.* **Systema evaluation on the HEPG2 unseen-perturbation setting ($n = 496$ perturbations).** ADAPERT leads with $P\Delta_{pert} = 0.394$, $P\Delta20_{pert} = 0.384$, and centroid accuracy $= 0.685$. TxPert follows at 0.359. MORPH shows strong centroid accuracy (0.681) but lower $P\Delta_{pert}$ (0.284). CPA and Matching-Mean produce negative $P\Delta_{pert}$, failing to capture perturbation-specific effects in this sparse-DEG dataset.

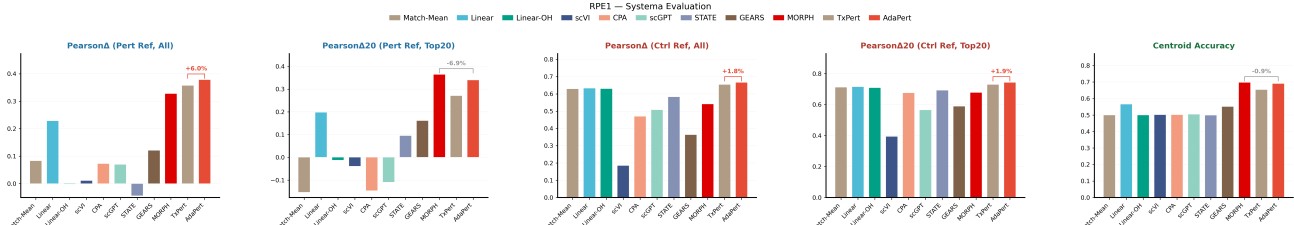

*Figure 17.* **Systema evaluation on the RPE1 unseen-perturbation setting ($n = 461$ perturbations).** ADAPERT achieves the best $P\Delta_{pert}$ (0.379) and the highest $P\Delta20_{ctrl}$ among perturbation-aware methods, with centroid accuracy of 0.691. TxPert follows at 0.357. MORPH achieves competitive $P\Delta20_{pert}$ (0.365) but lower perturbation-reference metrics. Linear baselines and foundation models (scGPT, scVI) lag substantially behind.

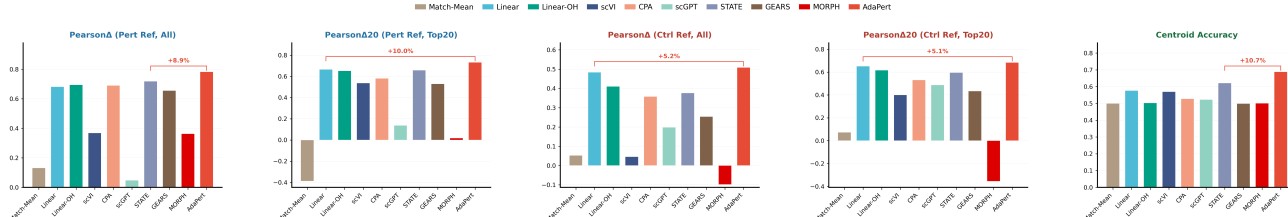

*Figure 18.* **Systema evaluation on the K562 unseen-cell-line setting ($n = 1,087$ perturbations; trained on RPE1+HEPG2+JURKAT, tested on K562).** ADAPERT achieves the highest $P\Delta_{pert}$ (0.783), $P\Delta20_{pert}$ (0.733), and centroid accuracy (0.689), outperforming the second-best STATE (0.718, 0.658, 0.622) by a substantial margin. This challenging cross-cell-line transfer setting demonstrates ADAPERT's ability to generalize perturbation predictions to entirely unseen cellular contexts with different gene-regulation programs.

*Table 14.* Systema evaluation metrics on K562 ($n = 272$). $P\Delta$: perturbation/control-referenced Pearson-$\Delta$ (All genes or top-20 DE). Best value per column in **bold**.

| Method | $P\Delta$ **(Pert)** | $P\Delta20$ **(Pert)** | $P\Delta$ **(Ctrl)** | $P\Delta20$ **(Ctrl)** | $N$ **Cond.** | **Centroid Acc.** |
|---|---|---|---|---|---|---|
| Matching-Mean | $-0.1190$ | $0.1870$ | $0.3865$ | $0.5401$ | 272 | $0.5000$ |
| Linear | $0.2842$ | $0.3027$ | $0.4614$ | $0.5866$ | 272 | $0.6203$ |
| Linear-OneHot | $0.0009$ | $-0.1237$ | $0.3870$ | $0.5467$ | 272 | $0.5000$ |
| scVI | $0.0426$ | $-0.1200$ | $-0.1188$ | $-0.1189$ | 272 | $0.5002$ |
| CPA | $0.0306$ | $-0.1478$ | $0.2713$ | $0.3855$ | 272 | $0.5013$ |
| scGPT | $0.0735$ | $0.0409$ | $0.3354$ | $0.5302$ | 272 | $0.5203$ |
| STATE | $-0.0539$ | $0.0692$ | $0.2208$ | $0.3746$ | 272 | $0.4890$ |
| GEARS | $0.0364$ | $0.3375$ | $0.2396$ | $0.5449$ | 272 | $0.5551$ |
| MORPH | $0.3624$ | $0.4419$ | $0.4408$ | $0.6269$ | 262 | $0.7292$ |
| TxPert | $0.4825$ | $0.5496$ | $0.5865$ | $0.7046$ | 272 | $0.7793$ |
| ADAPERT | $\mathbf{0.5338}$ | $\mathbf{0.5987}$ | $\mathbf{0.6184}$ | $\mathbf{0.7375}$ | 272 | $\mathbf{0.8105}$ |

*Table 15.* Systema evaluation metrics on JURKAT. Best value per column in **bold**.

| Method | $P\Delta$ **(Pert)** | $P\Delta20$ **(Pert)** | $P\Delta$ **(Ctrl)** | $P\Delta20$ **(Ctrl)** | $N$ **Cond.** | **Centroid Acc.** |
|---|---|---|---|---|---|---|
| Matching-Mean | $0.0942$ | $-0.1866$ | $0.3975$ | $0.5331$ | 448 | $0.5000$ |
| Linear | $0.1841$ | $0.2239$ | $0.4203$ | $0.5711$ | 448 | $0.5640$ |
| Linear-OneHot | $0.0034$ | $-0.0027$ | $0.3986$ | $0.5367$ | 448 | $0.5000$ |
| scVI | $0.0417$ | $-0.0729$ | $-0.1332$ | $-0.0760$ | 448 | $0.5032$ |
| CPA | $0.1018$ | $-0.1391$ | $0.2995$ | $0.4283$ | 448 | $0.5050$ |
| scGPT | $0.0857$ | $-0.0446$ | $0.1878$ | $0.3805$ | 448 | $0.5108$ |
| STATE | $0.0094$ | $-0.1039$ | $0.1758$ | $0.3405$ | 448 | $0.4954$ |
| GEARS | $0.0448$ | $0.1872$ | $0.2993$ | $0.4853$ | 446 | $0.5681$ |
| MORPH | $0.2662$ | $0.2886$ | $0.3091$ | $0.4980$ | 398 | $0.6763$ |
| TxPert | $0.3129$ | $0.3286$ | $0.4861$ | $0.6373$ | 446 | $0.6748$ |
| ADAPERT | $\mathbf{0.3450}$ | $\mathbf{0.3796}$ | $\mathbf{0.5003}$ | $\mathbf{0.6527}$ | 446 | $\mathbf{0.6867}$ |

*Table 16.* Systema evaluation metrics on HEPG2. Best value per column in **bold**.

| Method | P$\Delta$ (Pert) | P$\Delta$20 (Pert) | P$\Delta$ (Ctrl) | P$\Delta$20 (Ctrl) | $N$ Cond. | Centroid Acc. |
|---|---|---|---|---|---|---|
| Matching-Mean | $-0.1301$ | 0.0934 | 0.3850 | 0.5004 | 497 | 0.5000 |
| Linear | 0.2262 | 0.2388 | 0.4006 | 0.5253 | 497 | 0.5756 |
| Linear-OneHot | 0.0003 | 0.0077 | 0.3854 | 0.5017 | 497 | 0.5000 |
| scVI | 0.0132 | $-0.0399$ | $-0.0025$ | 0.2074 | 497 | 0.5045 |
| CPA | $-0.0871$ | 0.0753 | 0.3640 | 0.4867 | 497 | 0.5086 |
| scGPT | 0.0623 | 0.1266 | 0.3397 | 0.4946 | 497 | 0.5090 |
| STATE | 0.0194 | $-0.0045$ | 0.2475 | 0.4323 | 497 | 0.5009 |
| GEARS | 0.1488 | 0.2432 | 0.2346 | 0.4400 | 496 | 0.5965 |
| MORPH | 0.2836 | 0.3341 | 0.3464 | 0.5139 | 384 | 0.6805 |
| TxPert | 0.3591 | 0.3422 | 0.4502 | 0.5805 | 496 | 0.6698 |
| ADAPERT | **0.3943** | **0.3835** | **0.4694** | **0.6072** | 496 | **0.6849** |

*Table 17.* Systema evaluation metrics on RPE1. Best value per column in **bold**.

| Method | P$\Delta$ (Pert) | P$\Delta$20 (Pert) | P$\Delta$ (Ctrl) | P$\Delta$20 (Ctrl) | $N$ Cond. | Centroid Acc. |
|---|---|---|---|---|---|---|
| Matching-Mean | 0.0836 | $-0.1536$ | 0.6288 | 0.7122 | 464 | 0.5000 |
| Linear | 0.2287 | 0.1977 | 0.6332 | 0.7150 | 464 | 0.5654 |
| Linear-OneHot | $-0.0015$ | $-0.0122$ | 0.6302 | 0.7093 | 464 | 0.5000 |
| scVI | 0.0114 | $-0.0387$ | 0.1845 | 0.3944 | 461 | 0.5019 |
| CPA | 0.0731 | $-0.1465$ | 0.4701 | 0.6765 | 464 | 0.5022 |
| scGPT | 0.0701 | $-0.1091$ | 0.5077 | 0.5656 | 464 | 0.5043 |
| STATE | $-0.0438$ | 0.0954 | 0.5834 | 0.6927 | 461 | 0.4989 |
| GEARS | 0.1210 | 0.1613 | 0.3635 | 0.5894 | 461 | 0.5508 |
| MORPH | 0.3277 | **0.3647** | 0.5409 | 0.6784 | 405 | **0.6977** |
| TxPert | 0.3571 | 0.2709 | 0.6545 | 0.7297 | 461 | 0.6539 |
| ADAPERT | **0.3786** | 0.3396 | **0.6665** | **0.7434** | 461 | 0.6911 |

*Table 18.* Systema evaluation metrics on the K562 cross-cell-line setting (trained on RPE1+HEPG2+JURKAT, tested on K562). Best value per column in **bold**.

| Method | P$\Delta$ (Pert) | P$\Delta$20 (Pert) | P$\Delta$ (Ctrl) | P$\Delta$20 (Ctrl) | $N$ Cond. | Centroid Acc. |
|---|---|---|---|---|---|---|
| Matching-Mean | 0.1319 | $-0.3887$ | 0.0528 | 0.0725 | 1092 | 0.5000 |
| Linear | 0.6817 | 0.6664 | 0.4839 | 0.6511 | 1092 | 0.5762 |
| Linear-OneHot | 0.6947 | 0.6519 | 0.4102 | 0.6159 | 1092 | 0.5030 |
| scVI | 0.3678 | 0.5374 | 0.0457 | 0.4001 | 1092 | 0.5698 |
| CPA | 0.6895 | 0.5817 | 0.3577 | 0.5304 | 1092 | 0.5276 |
| scGPT | 0.0475 | 0.1380 | 0.1979 | 0.4870 | 1087 | 0.5225 |
| STATE | 0.7184 | 0.6579 | 0.3766 | 0.5953 | 1092 | 0.6217 |
| GEARS | 0.6548 | 0.5304 | 0.2540 | 0.4340 | 1087 | 0.4994 |
| MORPH | 0.3631 | 0.0179 | $-0.0983$ | $-0.3553$ | 1052 | 0.5007 |
| ADAPERT | **0.7826** | **0.7330** | **0.5091** | **0.6844** | 1087 | **0.6885** |

## D. DEG Sensitivity Analysis

The set of differentially expressed genes (DEGs) is used both as a training target and as an evaluation reference, so the chosen DEG-definition threshold could in principle bias our conclusions. We therefore test how sensitive ADAPERT is to this choice by sweeping the FDR cutoff and the |LFC| threshold and measuring, for each combination, both the number of resulting DEGs and prediction performance across six metrics on the K562, JURKAT, and RPE1 datasets (Figure 19).

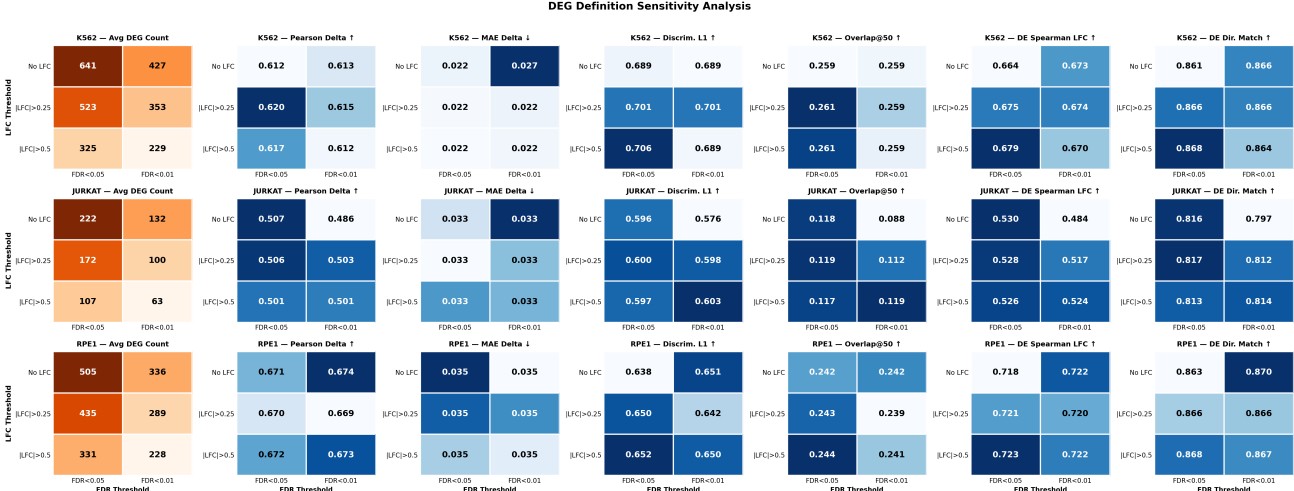

*Figure 19.* **Sensitivity of ADAPERT to DEG-definition thresholds across three cell lines (K562, JURKAT, RPE1).** Each row corresponds to a dataset. The first column (orange) shows the average number of DEGs per perturbation under each threshold combination. The remaining six columns (blue) show prediction performance across six evaluation metrics: Pearson-$\Delta$, MAE-$\Delta$, Discrimination L1, Overlap@50, DE Spearman LFC, and DE Direction Match. Stricter thresholds substantially reduce the number of DEGs used during training (e.g., K562: $641 \to 229$, JURKAT: $222 \to 63$, RPE1: $505 \to 228$) but have minimal impact on prediction performance across all six metrics. For instance, Pearson-$\Delta$ varies by less than $1.3\%$ in K562, $2.1\%$ in JURKAT, and $0.7\%$ in RPE1. FDR $< 0.05$ with $|LFC| > 0.25$ consistently yields the best or near-best performance, demonstrating that ADAPERT is robust to the specific DEG definition.

Prediction quality is stable across a wide range of DEG definitions: tightening the thresholds removes a large fraction of borderline DEGs from training yet leaves all six metrics essentially unchanged. This indicates that the gains reported elsewhere are not an artifact of a particular DEG cutoff, and it justifies our default choice of FDR $< 0.05$ with $|LFC| > 0.25$.

## E. Comprehensive Analysis on KG Variants

We conduct a comprehensive ablation on how the underlying biological knowledge graph (KG) affects ADAPERT, varying (i) the graph source, (ii) edge density, (iii) the context-extraction mechanism, and (iv) the attention threshold used for adaptive subgraph selection. All variants share the same architecture and training setup on the K562 dataset and differ only in the studied factor. Figures 20–24 report the results.

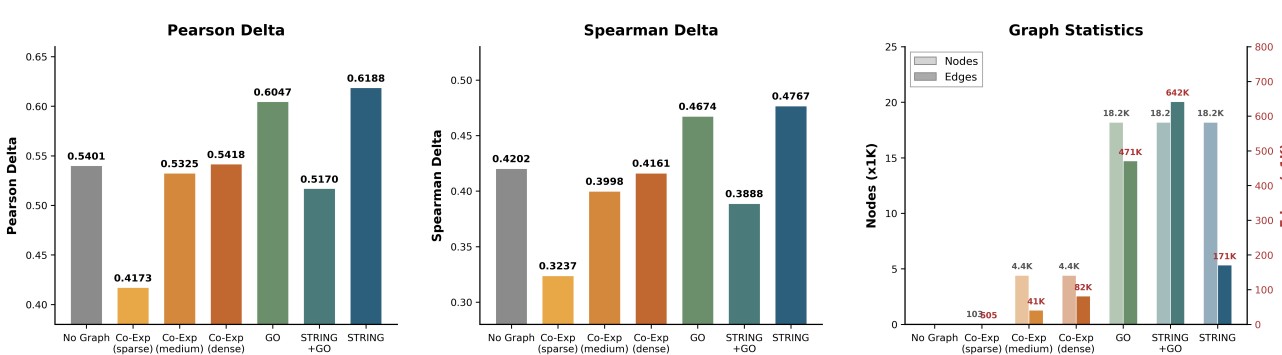

*Figure 20.* **Knowledge graph source comparison for perturbation prediction.** We test seven KG configurations using ADAPERT on the K562 single-cell dataset. All models share the same architecture and training setup, differing only in the graph source. Co-expression graphs are built from control cells ($n = 10,691$, 5,000 genes) using absolute Pearson correlation, following the GEARS protocol (Roohani et al., 2024). We test three sparsity levels: sparse (top-20, threshold $= 0.4$), medium (top-10, no threshold), and dense (top-20, no threshold). We compare co-expression graphs, a Gene Ontology (GO) graph, a STRING PPI graph, their combination (STRING+GO), and a no-graph baseline. **(Left, Middle)** Pearson and Spearman correlation of expression changes ($\Delta$) for unseen perturbations. The STRING PPI graph shows the best performance (Pearson-$\Delta = 0.6188$), followed by GO (0.6047). We find two main results. **First, node coverage is the key factor for perturbation embedding quality**: co-expression graphs cover only $\sim$5K highly variable genes, while curated KGs cover 18.2K nodes. Even when edge density increases from sparse to dense, co-expression graphs still fail to match STRING, which has $4\times$ more node coverage. **Second, with enough node coverage, moderate edge density better captures perturbation-specific signals**: STRING outperforms the combined STRING+GO graph even with $3\times$ fewer edges, suggesting that an overly dense graph adds noise and obscures useful information. Overall, the best KG for perturbation prediction needs broad gene coverage and selective, meaningful biological edges rather than merely high density. **(Right)** Graph statistics showing node coverage and edge counts for all configurations.

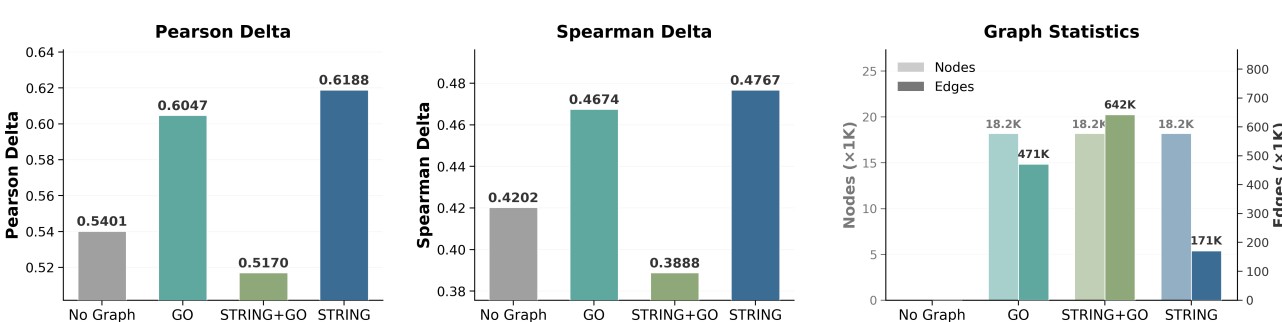

*Figure 21.* **Knowledge graph source comparison (without co-expression baselines).** The same experiment as Figure 20, showing only the four primary KG configurations: No Graph, GO, STRING+GO, and STRING.

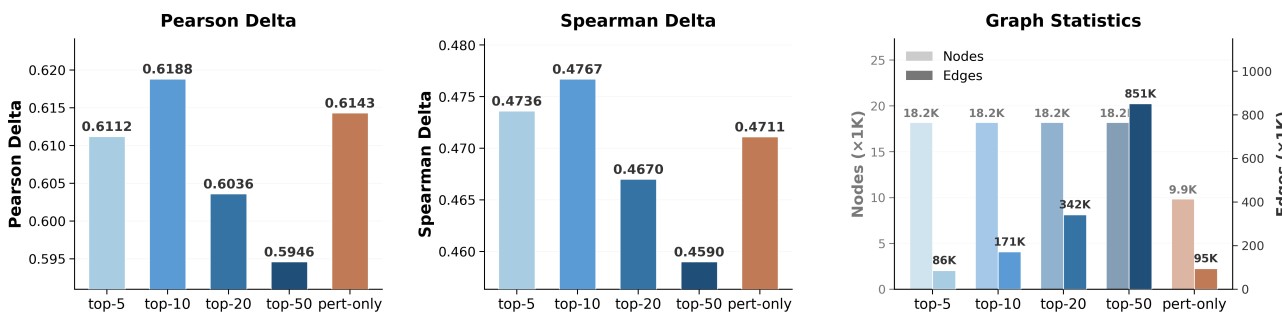

*Figure 22.* **Effect of edge density in the STRING PPI graph on perturbation prediction.** We test how graph sparsity affects prediction quality by varying the number of neighbors kept per gene (top-$K$ filtering) in the STRING network. All models use the same ADAPERT architecture on the K562 single-cell dataset, differing only in edge density. We also add a perturbation-only (pert-only) baseline, which builds a graph using only the perturbed genes and their direct neighbors. **(Left, Middle)** Pearson and Spearman correlation of predicted expression changes ($\Delta$) for unseen perturbations. The top-10 setup shows the best performance, and performance drops smoothly with both sparser (top-5) and denser (top-50) graphs: too few edges limit message passing, while too many edges add noise. The pert-only baseline beats both top-20 and top-50 despite having far fewer edges (95K vs. 342K/851K), indicating that **a focused subgraph around perturbed genes** is more useful than a globally dense but noisy graph. **(Right)** Graph statistics. All standard STRING setups share the same 18.2K nodes, with edge counts ranging from 86K (top-5) to 851K (top-50); the pert-only graph uses a smaller node set (9.9K) and 95K edges.

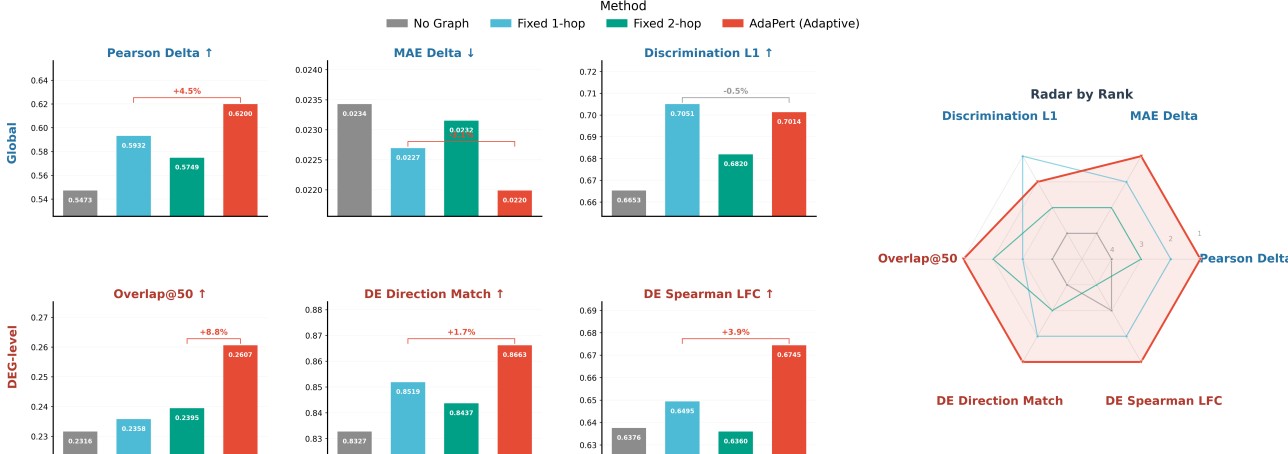

*Figure 23.* **Comparison of adaptive vs. fixed context subgraph methods.** We test four ways to extract context: No Graph (no context), Fixed 1-hop (sum of 1-hop neighbor embeddings), Fixed 2-hop (sum of 2-hop neighbor embeddings), and ADAPERT (adaptively learned context) on the K562 single-cell dataset. **(Top row) Global metrics.** ADAPERT shows the best Pearson-$\Delta$ and the lowest MAE-$\Delta$. **(Bottom row) DEG-level metrics.** ADAPERT brings steady gains across Overlap@50, DE Direction Match, and DE Spearman LFC. **(Right)** Radar chart ranking each method across all seven metrics; ADAPERT ranks first in 6 of 7 metrics. Learning an adaptive subgraph captures more useful context than using fixed neighbor connections.

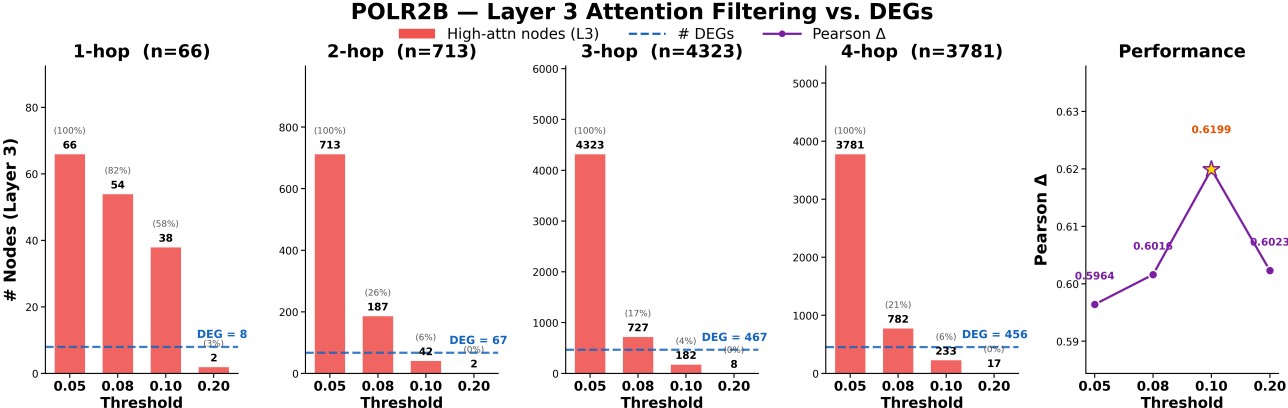

*Figure 24.* **Sensitivity analysis on the attention threshold** $T$. We examine how different attention thresholds ($T \in \{0.05, 0.08, 0.10, 0.20\}$) affect node selection across the perturbed gene's (POLR2B) $k$-hop STRING neighborhood ($k = 1$ to $4$). Red bars show the number of selected nodes (attention score $\alpha_i > T$), while blue dashed lines show the actual number of ground-truth DEGs. **(Panels 1–4)** At a low threshold ($T = 0.05$), the model keeps almost all neighbors (100%). At $T = 0.10$, the number of selected nodes is drastically reduced, closer to the scale of the actual DEG count at 1-hop (2 vs. 8) and 2-hop (42 vs. 67), indicating that the learned attention filters out topological noise. At larger distances (3-hop and 4-hop), $T = 0.05$ still retains thousands of nodes, whereas $T = 0.10$ narrows this to a much tighter subset (182 and 233 nodes, respectively). **(Rightmost) Performance across thresholds.** Pearson-$\Delta$ peaks at $T = 0.10$; an overly strict threshold ($T = 0.20$) drops too many useful nodes and performance falls. This shows that ADAPERT's attention can identify perturbation-relevant genes without direct DEG labels during training, and justifies our $[0.05, 0.20]$ threshold range for adaptive subgraph extraction.

## F. Analysis on Mean Collapse

A central failure mode in perturbation prediction is *mean collapse*, where a model attains high global correlation by predicting the average transcriptional response rather than perturbation-specific effects, thereby underestimating the largest gene-level changes. To assess whether ADAPERT mitigates this behavior, we compare prediction error separately on DEG and non-DEG genes against the strongest baselines (Figure 25).

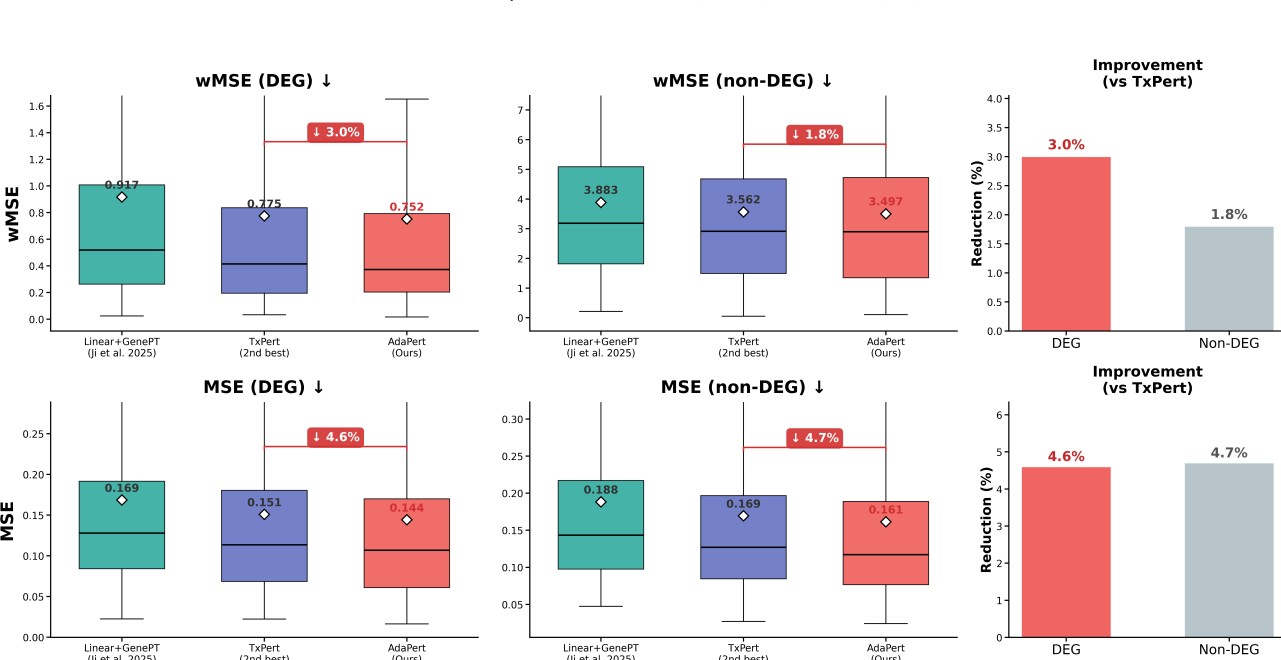

*Figure 25.* **DEG vs. non-DEG prediction error on K562** ($n = 272$ **perturbations**). Three methods are compared: a linear probe on GenePT embeddings (Linear+GenePT), TxPert (the second-best baseline), and ADAPERT. Top row: rank-weighted MSE (wMSE; weights inversely proportional to DEG rank by |LFC|, emphasizing top-impact genes). Bottom row: unweighted MSE. Left two columns: boxplots for DEG and non-DEG genes, respectively; diamonds denote means. Right column: normalized improvement of ADAPERT over TxPert. On wMSE, ADAPERT's improvement is larger on DEG genes (3.0%) than non-DEG genes (1.8%), confirming that the gains specifically target the highest-impact differentially expressed genes— the genes most susceptible to mean collapse. On unweighted MSE, improvements are comparable (DEG 4.6%, non-DEG 4.7%), indicating no degradation of non-DEG prediction quality. The consistent progression Linear → TxPert → ADAPERT across all four panels demonstrates that mean collapse is progressively mitigated by KG-based message passing and adaptive context extraction.

## G. Case Study

Beyond aggregate metrics, we examine whether ADAPERT's predictions translate into biologically meaningful, cell-type-specific signals at the pathway level. Using Gene Set Enrichment Analysis (GSEA) on the K562 dataset, we first quantify how well each method recovers significant pathways (Figure 26), and then analyze, pathway by pathway, how well each method ranks perturbations by their pathway-level effect (Figure 27).

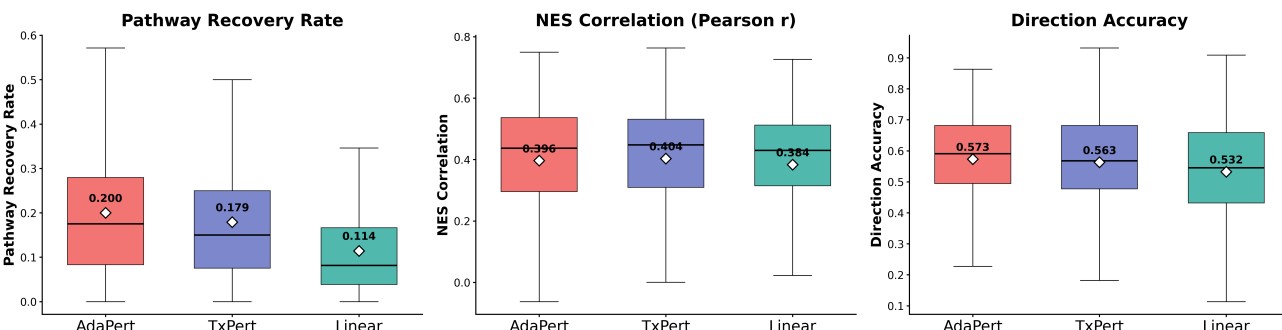

*Figure 26.* **Downstream pathway analysis (quantitative).** We evaluate pathway recovery using Gene Set Enrichment Analysis (GSEA) across 272 test perturbations on K562 single cells, over the complete set of MSigDB Hallmark 2020 pathways (300 permutations). **(Left) Pathway Recovery Rate**: the fraction of ground-truth significant pathways (FDR $< 0.25$) also identified as significant in the predictions. **(Center) NES Correlation**: the Pearson $r$ between predicted and ground-truth Normalized Enrichment Scores (NES) across matched pathways. **(Right) Enrichment Direction Accuracy**: the fraction of pathways where the model correctly predicts the enrichment direction (up or down). Diamonds denote means; boxes span the interquartile range (IQR). ADAPERT shows the best recovery rate and the highest direction accuracy.

At the pathway level, ADAPERT recovers a larger fraction of significant Hallmark pathways than the baselines and predicts their enrichment direction more accurately. Resolving this to individual pathways reveals where the gains come from.

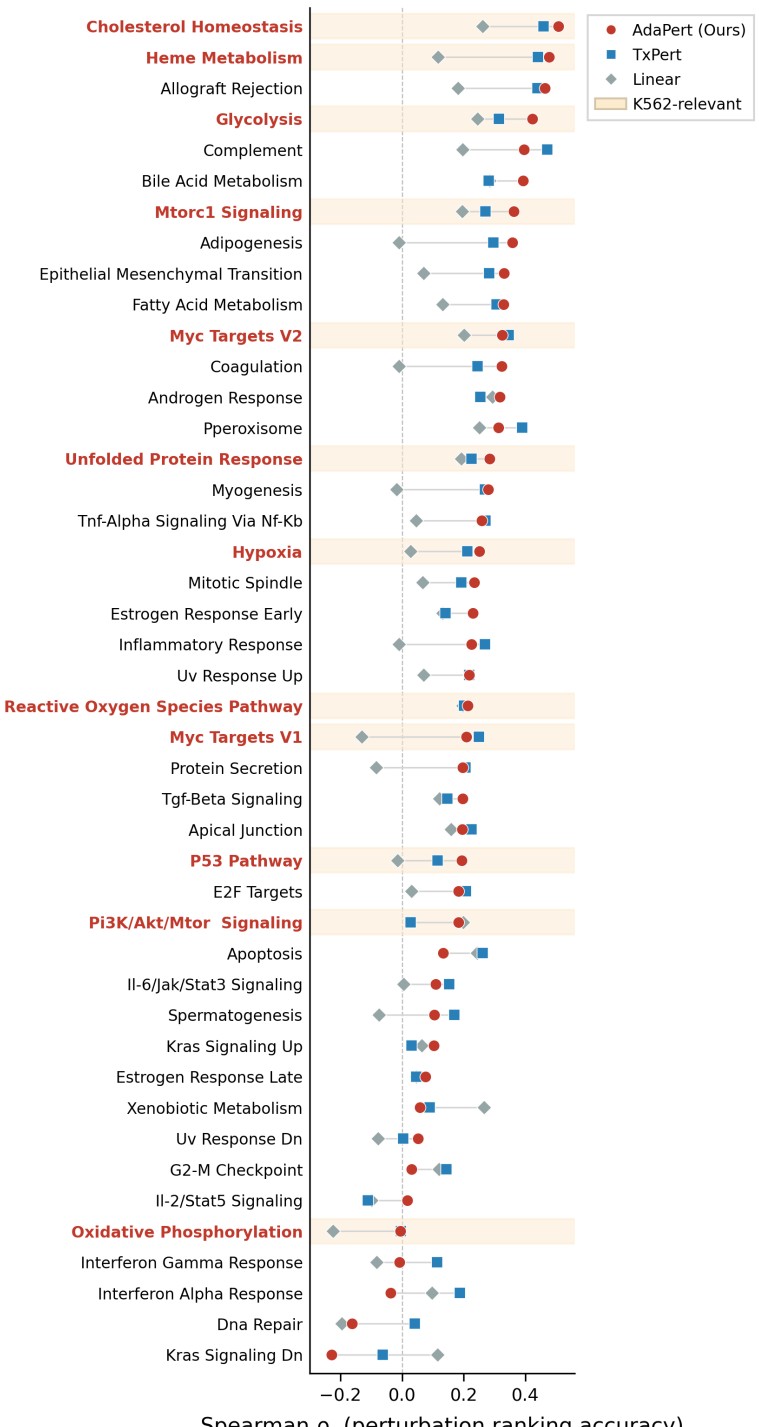

*Figure 27.* **Perturbation ranking accuracy across all 50 MSigDB Hallmark pathways on K562.** For each pathway, we compute the ground-truth NES from GSEA on real expression profiles, repeat the analysis on predicted profiles, and measure how well each method ranks perturbations by pathway effect using Spearman $\rho$. Pathways are sorted by ADAPERT's $\rho$ (descending). Three methods are compared: ADAPERT (ours), TxPert, and Linear-GenePT, on 272 unseen test perturbations. K562 is a chronic myeloid leukemia (CML)/erythroleukemia cell line widely used in large-scale Perturb-seq studies; pathways with established biological relevance to K562 are highlighted (orange background, bold labels), including heme metabolism (a defining feature of erythroleukemia), glycolysis (Warburg effect), mTORC1 and PI3K/AKT/mTOR signaling (downstream of the BCR-ABL oncogene), cholesterol homeostasis, the P53 pathway, and the unfolded protein response. ADAPERT outperforms TxPert on **9 of 12 K562-relevant pathways**, with the strongest advantages on core metabolic pathways (Cholesterol Homeostasis, Heme Metabolism, Glycolysis, and mTORC1 Signaling). A few pathways (e.g., DNA Repair, KRAS Signaling Dn) show negative correlations across all methods, indicating pathway-level signals that remain challenging for current approaches.

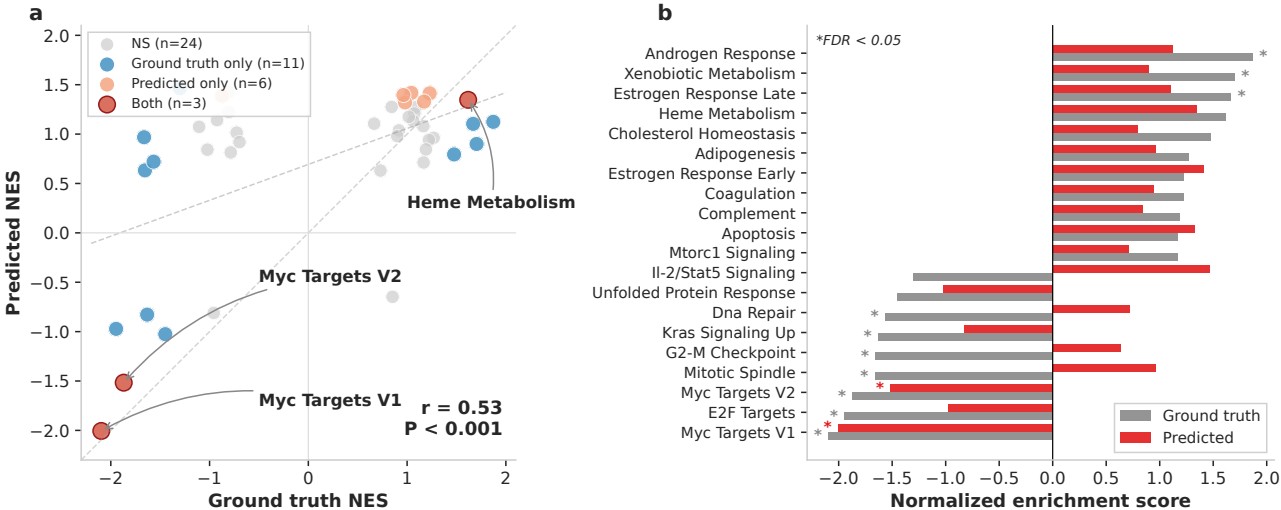

*Figure 28.* Correlation of pathway enrichment between predicted and ground truth responses for HIRA knockdown. Each point represents one of 44 Hallmark pathways, with the x-axis showing ground truth NES and the y-axis showing predicted NES. Colors indicate significance status (FDR < 0.25): gray, non-significant in both; blue, significant in ground truth only; coral, significant in predictions only; red, significant in both. Pathways significant in both analyses (Myc Targets V1, Myc Targets V2, and Heme Metabolism) are labeled. Dashed lines indicate the diagonal ($y = x$) and the linear regression fit. Pearson correlation $r = 0.53$, $P < 0.001$.

The improvements concentrate on pathways with established biological relevance to K562, indicating that ADAPERT preferentially captures cell-type-specific biological programs rather than generic transcriptional shifts.

### G.1. Pathway Enrichment Correlation for HIRA Knockdown

To assess agreement between predicted and experimental pathway enrichment, we compare normalized enrichment scores (NES) across all 44 Hallmark gene sets. As shown in Figure 28, predicted and ground truth NES values show a significant positive correlation ($r = 0.53$, $P < 0.001$).

Among the 14 pathways significantly enriched in the ground truth analysis (FDR < 0.25), the model identifies 9 as significant. Three pathways (Myc Targets V1, Myc Targets V2, and Heme Metabolism) are significant in both analyses. Agreement is strongest for pathways with large effect sizes. For example, Myc Targets V1 shows closely matched enrichment between prediction (NES = −2.01) and ground truth (NES = −2.10), indicating accurate recovery of both direction and magnitude of pathway-level effects.

## H. Additional Related Works

### H.1. Data-driven and general-purpose modeling approaches

A broad class of prior work models transcriptional responses to perturbations primarily through *data-driven learning*, without explicitly encoding biological mechanisms. Early generative frameworks such as (Lopez et al., 2018; Lotfollahi et al., 2019) learn latent representations of gene expression and infer perturbation effects through shifts in latent space. Subsequent methods, including (Lotfollahi et al., 2023; Adduri et al., 2025), extend this paradigm by conditioning latent variables on perturbation identities and cellular contexts.

Related to these approaches, several models formulate perturbation prediction as a *distributional mapping problem*. Optimal-transport–based methods such as (Bunne et al., 2023; Chen et al., 2025) and causal transport models like (Dong et al., 2023) aim to align control and perturbed cell populations at the distribution level. While effective at capturing global expression shifts, these methods are not explicitly designed to recover sparse gene-level effects.

More recently, large-scale *foundation models* have been introduced for single-cell biology, including (Cui et al., 2024; Hao et al., 2024; Theodoris et al., 2023; Rosen et al., 2023; Pearce et al., 2025). These models learn transferable gene or cell representations from massive datasets and are often used as pretrained encoders for downstream tasks. However, they do not

explicitly model perturbation-specific sparsity or directionality, and their predictions may still be dominated by averaged transcriptional responses.

## H.2. Knowledge-driven perturbation models

To address the limitations of purely data-driven approaches, a growing line of work incorporates *biological prior knowledge* into perturbation response modeling. Methods such as (Roohani et al., 2024; Wenkel et al., 2025) leverage gene–gene interaction networks or pathway graphs to propagate perturbation signals through known biological relationships, improving generalization to unseen perturbations.

Recent studies further explore the integration of *textual and semantic biological knowledge*. Approaches including (Chen & Zou, 2024; Istrate et al., 2024; Wu et al.; Istrate et al., 2025) use pretrained language models to construct gene representations from literature, functional annotations, or structured biological descriptions. These methods demonstrate that external knowledge can complement expression data, particularly in low-data or out-of-distribution settings.

*However, most existing knowledge-driven models treat biological knowledge as static and globally shared across perturbations.* Dense graphs or fixed embeddings are typically reused for all perturbations, which can propagate irrelevant interactions and obscure perturbation-specific signals. *This static usage of knowledge limits the ability of models to adaptively focus on the most relevant biological substructures for a given genetic intervention.*

In addition, prior knowledge is often integrated uniformly, without explicit mechanisms to separate true perturbation-induced signals from background transcriptional variation.

## H.3. Positioning of this work

Our work builds on the knowledge-driven paradigm by introducing *perturbation-conditioned adaptation* in the use of biological knowledge. Rather than relying on static graphs or fixed embeddings, we learn sparse, perturbation-specific subgraphs that dynamically emphasize relevant biological interactions. This design complements prior data-driven and knowledge-based approaches and enables more accurate recovery of perturbation-specific transcriptional signals.

