# OpenReview forum: "Learning Adaptive Perturbation-Conditioned Contexts for Robust Transcriptional Response Prediction"
_ICML.cc/2026/Conference — ICML 2026 regular_

### Official Review · Reviewer_t8Qj · 2026-03-08

**Soundness:** 3
**Presentation:** 3
**Significance:** 2
**Originality:** 2
**Overall Recommendation:** 4
**Confidence:** 4

**Summary:**

ADAPERT predicts perturbed gene expression by combining a control-cell encoder-decoder with a perturbation-conditioned context vector learned from a biological knowledge graph. For each perturbation, it selects a sparse, perturbation-specific subgraph using graph structure plus an LM embedding of the perturbed gene’s text description. Training adds non-DEG suppression (Huber on predicted deltas) and alignment to DEG-only response targets to reduce mean-collapse and better recover perturbation-specific signals.

**Compliance With Llm Reviewing Policy:**

Affirmed.

**Final Justification:**

My concerns are addressed to some extent. The additional experimental results look satisfactory.

**Key Questions For Authors:**

1. The authors first define a generic perturbation embedding $z_p$ (Eq.2) and later introduce the perturbation-conditioned context vector $z_{\text{context}}^{(p)}$ (Eq.13). Is $z_p$ fully replaced by $z_{\text{context}}^{(p)}$ in the decoder for ADAPERT, or are both used/combined?
2. The paper defines the encoder and decoder functionally ($ENC_\theta$, $DEC_\phi$) and GNN but does not specify their exact architectures. Could you clarify the detailed layer structure of the encoder and decoder (e.g., number of layers, hidden dimensions, activation functions, normalization, dropout) for the best-performing case?
3. In Eq.(13), the perturbation-conditioned context representation is computed using a summation over selected node embeddings rather than an average. Since the number of selected nodes may vary across perturbations, this makes the magnitude of $z_{\text{context}}^{(p)}$ dependent on subgraph size. Also, it uses summation over $h_v$. Would it make sense to use $c_v$ instead of $h_v$?
4. Can LM encoding be used for knowledge graph nodes instead of one-hot encoding?

**Limitations:**

I don't see the limitations are discussed.

**Strengths And Weaknesses:**

## Strengths

- The overall performance comparison experiment is well designed with appropriate baselines and DEG-aware metrics.
- The writing and the demo clearly connects empirical results to the mean-collapse hypothesis, making the paper easy to follow and technically coherent.

## Weaknesses

- The technical novelty is incremental. The adopted techniques such as the integration of knowledge graphs, huber loss, and cosine loss are all standard and well-understood methods. The authors seem simply put them together as a pipeline and lead to a good overall performance.

- A rigorous ablation study is missing. A more extensive ablation study to evaluate the contributions of every component would provide insight on the proposed method. For example, Figure 4 shows with\without non-DEG loss, and the performance gains do not seem significant in most cases. In some cases, removing components appears to perform even better than the full model. It seems suggest that $L_{\rm non}$ loss is not contributing. Although Figure 5 shows the tuning of the coefficient of the non-DEG loss affecting performance, the performance change in the interval 0 to 0.01 is insignificant. It seems consistent with Figure 4 that the proposed loss's contribution to the performance gain is minimal.

- The down-stream task in Figure 6 is not informative. The authors claim significant positive correlations. However, the figure show in many cases the scores between model predictions and ground-truths are mismatched, even with opposite signs. Furthermore, Since there are no other baseline methods (such as in Table 1 and 2) presented for comparison, it's not possible to evaluate how well the proposed method performs.

- While the overall performance are promising, the experimental section lacks some reproducibility details (e.g., exact encoder/decoder architectures, number of layers, hyperparameter selection protocol, etc.).

- The paper says experiments on “HEPG2.Replogle” earlier, but the tables show RPE1.Replogle. That inconsistency needs fixing.
- Typo: "Subraph" in line 187.

---

> ### Author Rebuttal · Authors · 2026-03-31
>
> We thank the reviewer for acknowledging that the experiments are "well designed with appropriate baselines and DEG-aware metrics" and that the writing "clearly connects empirical results to mean-collapse hypothesis." We address each concern below.
>
> **[W1&W2]** We appreciate this question and welcome the chance to clarify. AdaPert was driven by a specific observation: when models optimize reconstruction across all ~5,000 genes equally, ~90–95% of non-responsive genes dominate the loss, pulling predictions toward the mean. This issue has been increasingly noted, but few methods tackle it at the training objective level. Our components each address a different angle: the adaptive subgraph extraction selects perturbation-relevant KG neighbors (`Appendix Figure 9`), ground-truth DEGs spread across distant hops, so fixed neighborhoods miss them while learned attention can reach them (`Figure 4` in https://anonymous.4open.science/api/repo/rebuttal-AF11/file/index.html#nSvh_kg). The non-DEG guidance loss regularizes genes expected to remain near baseline, redirecting capacity toward genes that actually change.
> On the ablation: the key insight from original Figure 4 is that contributions depend on perturbation strength. The non-DEG loss matters most for weak perturbations where false positives dominate; the context module matters most for strong perturbations where cascading effects need capturing. Stratified ablations with well-calibrated metrics (following Reviewer #nSvh, https://anonymous.4open.science/api/repo/rebuttal-AF11/file/index.html#nSvh_collapse) confirm this: each component contributes most in the regime it was designed for. Our contribution is diagnosing mean-collapse as the core bottleneck and a training framework addressing it from complementary directions.
>
> **[W3]** We thank the reviewer for this suggestion. The `original Figure 6` showed AdaPert can recover pathway-level trends for HIRA, but a single case without baselines limits interpretability. We now provide a full downstream evaluation across all 272 K562 test perturbations (details in Reviewer #nSvh Q4 and in https://anonymous.4open.science/api/repo/rebuttal-AF11/file/index.html#t8Qj_downstream), comparing AdaPert against TxPert and Linear on **DEG identification** (Figure 1), **pathway recovery** (Figure 2), and **target prioritization** (Figure 3). The result shows AdaPert outperforms baselines on all three tasks.
>
> **[W4]** Hyperparameters for all experiments were reported in the (`Appendix Table 9`). We additionally clarify the subgraph attention threshold $T$, which was selected from {0.05, 0.08, 0.10, 0.2} via validation performance.
> **[Q1]** Thank you for raising this point. In AdaPert, the perturbation embedding `z_p = [z_context, \hat{s}_p]` is formed by concatenating the perturbation-conditioned context vector `z_context` with the projected semantic embedding `\hat{s}_p` (Eq. 9). We will clarify this in the revised manuscript.
> **[Q2]** For the best-performing case in K562, the encoder is a 2-layer MLP (5000→512→128) with BatchNorm and LeakyReLU. The GNN consists of 4 GATv2 layers with skip-cat connections, projecting to a 128-dim output. The decoder is a 2-layer MLP (3328→256→5000) with BatchNorm, LeakyReLU, and Dropout(0.2), where the input dimension 3328 = 128 (basal) + 128 (GNN) + 3072 (LLM embedding).
>
> **[W5]** We thank the reviewer for catching this. Line 297 should read "RPE1.Replogle". We have now also added HepG2.Replogle results in the expanded evaluation in https://anonymous.4open.science/api/repo/rebuttal-AF11/file/index.html#nSvh_comprehensive `(Figures 1-5)`, so both cell lines are included in the revised manuscript. [W6] Thank you for spotting this. We corrected it in the revision.
>
> **[Q3]** Thank you for this question. We chose summation because different perturbations have different effect sizes, and sum aggregation naturally encodes this difference. Averaging would lose this signal. Summation on `h_v` vs `c_v`: we tried aggregating `c_v = [h_v, s_p_hat]`, but we find LLM embedding (`dim=3072`) dominates over the GNN embedding (`dim=128`). Since `\hat{s}_p` is fixed per gene, summing it across neighbors just scales a static vector, degrading structural differences. Using `h_v` alone may keep the context grounded in graph-propagated features that differentiate selected nodes.
>
> **[Q4]** Thank you for this question. We tested this and found that `one-hot initialization` outperforms `LM initialization` across all metrics in https://anonymous.4open.science/api/repo/rebuttal-AF11/file/index.html#t8Qj_lm_init. We attribute this to an imbalance in coverage: since LM descriptions exist only for perturbation genes, nearly half the KG nodes have no LM embedding and must be randomly initialized. This gap between semantically-rich and randomly-initialized nodes disrupts GNN message passing. Starting all nodes from one-hot avoids this issue, allowing the GNN to build representations on equal footing across the entire graph.

---

> > ### Author Rebuttal · Reviewer_t8Qj · 2026-04-04
> >
> > I appreciate the authors' rebuttal. The added ablation study looks convincing to me. the response to other concerns are satisfactory to some extent. I adjusted my score.

---

> > > ### Author Response · Authors · 2026-04-05
> > >
> > > We sincerely thank the reviewer for the positive review and the constructive suggestions throughout the review process, which led to a more comprehensive evaluation (expanded downstream analysis and ablation on LM initialization) and improved clarity of our manuscript. All rebuttal experiments and a Limitations section will be included in the revised version.

---

### Official Review · Reviewer_BXkW · 2026-03-12

**Soundness:** 3
**Presentation:** 3
**Significance:** 3
**Originality:** 3
**Overall Recommendation:** 4
**Confidence:** 5

**Summary:**

This study proposes a computational framework ADAPERT, for predicting transcriptional responses to genetic perturbations that addresses the mean-collapse problem. The work may have been a laborious one that I should appreciate the authors' time and patience to come up with some results.

**Compliance With Llm Reviewing Policy:**

Affirmed.

**Final Justification:**

The authors have addressed all my concerns. I have updated the scores.

**Key Questions For Authors:**

1. You should add comparison with recent single-cell foundation models like scGPT, scBERT, or Geneformer that might capture perturbation responses through different mechanisms.
2. The method relies exclusively on the STRING PPI network. It's unclear how sensitive performance is to this choice or whether integrating multiple knowledge sources (e.g., pathway databases, co-expression networks) could further improve results.
3. The knowledge graph inevitably misses some edges and may include false positives. How robust is ADAPERT to incomplete or noisy graph structure? Have the authors experimented with graph perturbation or edge dropout during training?
4. The method uses GPT-4o embeddings of NCBI descriptions. How sensitive are results to the choice of language model? Would fine-tuning these embeddings on perturbation data improve performance, or is a frozen embedding sufficient?

**Limitations:**

No.

**Strengths And Weaknesses:**

The identification of mean-collapse as a key failure mode is convincing. Combining structural graph information with semantic information (language model embeddings from NCBI descriptions) is effective and biologically meaningful. While the paper compares with several graph-based and VAE-based methods, it lacks comparison with recent single-cell foundation models like scGPT, scBERT, or Geneformer that might capture perturbation responses through different mechanisms. The method relies exclusively on the STRING PPI network. It's unclear how sensitive performance is to this choice or whether integrating multiple knowledge sources (e.g., pathway databases, co-expression networks) could further improve results.

---

> ### Author Rebuttal · Authors · 2026-03-31
>
> We thank the reviewer for recognizing that the identification of mean-collapse is "convincing" and that combining structural graph with semantic information is "effective and biologically meaningful." We address each concern below.
>
> **[Q1]** We thank the reviewer for this suggestion. Within the rebuttal period, we added the following representative baselines: scGPT (Nature Methods 2024) and Linear with one-hot encoding (Nature Methods 2025), as well as an advanced version with GenePT embeddings (Chen et al, 2024). These were evaluated alongside existing baselines across four cell lines (K562, RPE1, HEPG2, JURKAT) in the unseen perturbation setting, one cross-cell-line setting (K562-cross), and the SYSTEMA benchmark framework. Please refer to https://anonymous.4open.science/api/repo/rebuttal-AF11/file/index.html#BXkW_comprehensive. AdaPert consistently outperforms scGPT by a large margin across all settings, as well as both linear baselines. Full results and per-dataset details are provided in our response to `Reviewer nSvh (W4, W5, W7, Q1)` and `Reviewer mrYu (W3, Q2)`.
>
> **[Q2&Q3]** We thank the reviewer for these constructive questions on knowledge graph sensitivity and robustness. To address both concerns, we conducted a series of controlled experiments in https://anonymous.4open.science/api/repo/rebuttal-AF11/file/index.html#BXkW_kg.
> 1. *Graph source (Figure 1)*. We compared STRING PPI, Gene Ontology (GO), their combination, and no graph. STRING alone performs best, while GO also provides substantial gains over no-graph. Combining STRING+GO degrades performance, suggesting denser but noisier edges dilute perturbation-relevant signal. AdaPert benefits from curated, interaction-specific topology rather than simply more edges.
> 2. *Edge density (Figure 2)*. Within STRING, we varied neighbors retained per node (top-5 through top-50) and tested a pert-gene-only subgraph. Performance remains stable across a wide range of densities, demonstrating robustness to graph sparsification and incomplete coverage.
> 3. *Context aggregation (Figure 3)*. We compared no context, fixed 1-hop/2-hop aggregation, and adaptive extraction. Fixed 1-hop provides clear gains, but 2-hop dilutes signal. Adaptive extraction outperforms all fixed strategies, confirming learned perturbation-specific selection is more effective and implicitly robust to noisy edges (refer to `Reviewer nSvh W8`).
> 4. *Attention threshold analysis (Figure 4, Reviewer nSvh's suggestion)*. We visualized how node scoring filters the local KG neighborhood for a representative perturbation (POLR2B). At the optimal threshold, the number of high-attention nodes closely matches the ground-truth DEG count at each hop distance, suggesting the model learns to identify perturbation-relevant substructure. This provides interpretable evidence that the adaptive mechanism performs meaningful, biologically grounded selection rather than arbitrary filtering.
>
> **[Q4]** We thank the reviewer for this thoughtful question. Understanding the role of each component in the pipeline is important for both reproducibility and future development. We ran a controlled experiment swapping `GPT-4o` for `Claude Haiku 4.5` in the gene summary generation step, keeping the downstream embedding model (text-embedding-3-large, 3072-dim) identical https://anonymous.4open.science/api/repo/rebuttal-AF11/file/index.html##BXkW_lm.
> The setup exploits a natural variable: when NCBI + UniProt descriptions are available, 99.4% of summaries come from curated text and only 0.6% require LLM generation. With NCBI alone, 7.5% of genes lack substantive descriptions and the LLM must generate from scratch (`Figure 1, right`). Both LLMs receive the same reference text for well-annotated genes, and the difference only matters for the small tail of poorly annotated ones.
> When reference coverage is high (NCBI + UniProt), `GPT-4o` and `Claude` achieve nearly identical Pearson-Delta (0.6162 vs 0.6139, gap = 0.002), confirming robustness to LLM choice when summaries are grounded in curated data. With NCBI only, where more summaries must be generated de novo, Claude edges ahead (0.6104 vs 0.6054, gap = 0.005). Notably, the gap between description sources (NCBI+UniProt vs NCBI-only: \~0.01) is much larger than the gap between LLMs within the same source (~0.002–0.005). This tells us that what matters is the quality of the input descriptions — particularly whether protein-level annotations are included — not which LLM formats them. And we think exploring domain-adapted embeddings and richer curated sources is a promising direction for future work.

---

> > ### Author Rebuttal · Reviewer_BXkW · 2026-04-03
> >
> > Thank you for the detailed rebuttal. However, my concerns remain partially resolved. The authors did fully follow my suggestions.
> > Q1: I would still like to see the comparison performance on scBERT and Geneformer.
> > Q2：Could you test the sensitive performance for co-expression networks as the knowledge source?

---

> > > ### Author Response · Authors · 2026-04-05
> > >
> > > We thank the reviewer for the follow-up questions. Results for both are below.
> > >
> > > **Q1: I would still like to see the comparison performance on scBERT and Geneformer.**
> > >
> > > Following the reviewer's request, we evaluated Geneformer (Theodoris et al., Nature 2023), a 104M-parameter Transformer pretrained on ~30M single-cell profiles. We followed the original in silico perturbation protocol: the perturbed gene token is removed from the rank-value-encoded control cell, and a linear probe maps the [CLS] embedding to the predicted expression change. Please refer to the results in `Figures 1–4` via https://anonymous.4open.science/api/repo/rebuttal-AF11/file/index.html#BXkW_comprehensive.  The result shows Geneformer falls behind AdaPert across datasets, which is consistent with what we observe across foundation models: they tend to predict toward the global mean rather than capturing perturbation-specific changes, which is the core mean-collapse problem that AdaPert is designed to address.
> > >
> > > Regarding scBERT: we tried to include it, but unfortunately, the pretrained weights and training data are no longer available (confirmed by `scBERT GitHub Issues #63, #68..`). To still address your question, we evaluated AIDO.Cell (genbio-ai) as an alternative, a 100M-parameter Transformer with publicly available weights, using the same pretrained embedding + linear-probe protocol. Interestingly, despite different pretraining corpora (CELLxGENE census vs. Genecorpus-30M), architectures, and embedding dimensions, AIDO.Cell achieves Pearson Δ of 0.388, very close to Geneformer (0.386), both well below AdaPert. This suggests that the gap is likely not model-specific but reflects a shared limitation of general-purpose pretrained representations for this task.
> > >
> > >
> > > **Q2：Could you test the sensitive performance for co-expression networks as the knowledge source?**
> > >
> > > Thank you for this suggestion, and we share our results and some thoughts below. We first constructed co-expression graphs from control cells (n=10,691, 5,000 genes) using absolute Pearson correlation following the GEARS protocol, and tested three sparsity levels: sparse (top-20, threshold=0.4), medium (top-10, no threshold), and dense (top-20, no threshold). Full results are in `Figure 1a` via https://anonymous.4open.science/api/repo/rebuttal-AF11/file/index.html#BXkW_kg.
> > >
> > > We find all three co-expression variants fall below both curated KGs (STRING, GO). One likely reason is the difference in node coverage: co-expression graphs only cover ~5K highly variable genes, while curated KGs like STRING cover 18.2K nodes. **From a biological perspective**, co-expression reflects statistical associations in steady-state expression, while perturbation effects tend to propagate through mechanistic interactions like protein-protein binding and signaling cascades, which may explain why curated interaction networks are more informative here. This observation also touches on a broader question that motivated AdaPert's design: how to learn perturbation-specific patterns from biological knowledge graphs. Interestingly, our experiments suggest that naively combining multiple KG sources (e.g., STRING+GO) does not always help and can even add noise. Finding better ways to integrate diverse biological knowledge sources seems like a promising direction worth exploring.
> > >
> > > We sincerely thank the reviewer again for the constructive and thorough suggestions. These experiments have enriched both our empirical analysis and discussion, and we will incorporate all results into the revised manuscript.

---

### Official Review · Reviewer_nSvh · 2026-03-12

**Soundness:** 2
**Presentation:** 3
**Significance:** 2
**Originality:** 3
**Overall Recommendation:** 3
**Confidence:** 4

**Summary:**

The authors present AdaPert, a method for predicting transcriptional changes in response to genetic perturbations in single-cell RNA-seq data. They highlight mean-collapse as a key limitation of existing methods, where predicted gene expression shifts toward the global average rather than reflecting perturbation-specific responses, leading to underestimation of perturbation-responsive genes. To address this issue, the authors incorporate knowledge graphs into their model, seeking to separate true signals (response genes) from noise and better capture the biological structure of perturbation responses. They motivate their approach by the observation that only a small subset of genes exhibit perturbation-induced changes in response to typical genetic perturbations. AdaPert adaptively learns perturbation-specific knowledge graphs using differentially expressed genes between control and perturbed cells. The method is trained and tested using two CRISPR perturbation datasets and benchmarked against six other methods. The authors further analyze the contribution of their proposed loss function across different perturbation scenarios and assess biological relevance of the predictions.

**Compliance With Llm Reviewing Policy:**

Affirmed.

**Key Questions For Authors:**

1. Based on the “Data Statistics” section, the authors state that the evaluation is performed on the K562.Replogle and RPE1.Replogle datasets (line 564), while the main text also mentions HEPG2.Replogle (line 297). Could the authors clarify which datasets were used? Would it be possible to include additional dataset for validation to show AdaPert’s performance on unseen cell lines?
2. It seems that removing z-context and non-DEG loss has overall a rather moderate effect on model performance. Perhaps reporting median values would be more helpful to assess the effect size of the changes (Figure 4). It may be worth highlighting the model’s capacity to better discriminate small perturbations (Figure 4, top right), with overall best performance of the model for large perturbations.
3. In the “Introduction” the authors argue that “mean-collapse arises not from insufficient data or features but from a mismatch from the sparse nature of perturbation responses”, but they don’t explore this hypothesis. To better highlight the significance of their approach, it would be beneficial to include an experiment supporting this claim.
4. The authors do not demonstrate whether AdaPert would substantially improve downstream analyses, such as target selection, compared to simpler baselines. Showing this would help establish the practical utility of the method beyond improvements in evaluation metrics.

**Limitations:**

No. The paper lacks a discussion of limitations and focuses mainly on the positive results. For instance, issues such as generalization across cell types or donors and the impact of DEG selection on performance are not addressed.

The authors claim that their design enables robust modelling of perturbation changes under noisy settings, but they do not test it under severe noise or at least report the extent of noise in the input data.

**Strengths And Weaknesses:**

Strengths:
* The paper is well written and easy to follow, clearly explaining the motivation for perturbation-conditioned contexts, the role of the graph and text embeddings, and the used loss components. The authors clearly state the main contribution which is the context-dependent adaptive subgraph learning.
* The work is technically sound, with appropriate modeling choices (using graph and text-based embeddings, perturbation-conditioning and multi-term loss), standard optimization and training setup. The paper is based on empirical results with generally well-designed experiments and validations on unseen perturbations. The authors report multiple metrics, including DEG-specific ones, and provide ablations for graph context, non-DEG loss and subgraph alignment, which address the mean-collapse as the motivation of the presented work.
* The authors evaluate the effect of the non-DEG loss regularization on model performance across different perturbation groups (Figure 5), which helps assess its role in recovering signals under varying levels of noise.
* Using pathway enrichment scores and correlating them with the ground truth activity is a well-motivated and useful evaluation approach to assess overall biological activity based on predicted expression in perturbed cells.
* The paper addresses a rapidly evolving and highly competitive field, focusing on an important aspect of perturbation modeling. The use of knowledge graphs and the focus on DEG genes through the loss function is a biologically intuitive and reasonable design choice. Although the presented performance improvements are relatively modest, the authors suggest a promising direction for future research, with the proposed ideas likely to motivate future work in single-cell perturbation studies.

Weaknesses:
* The authors do not explore the dependency of the results on the chosen biological knowledge graphs, which introduces a major source of variation in itself.
* The description how the selected genes were identified should be clarified. For example, it is not clear if those genes were selected based on expression change > 0.1 or with p-value < 0.05 (lines 263 and 713). Which statistical tests/packages were used to identify them?
* The authors do not assess how sensitive AdaPert’s performance is to variation in the selection of these genes.
* The comparison with other methods is appreciated, but comparison with simpler linear models or mean baseline (as in the STATE paper https://doi.org/10.1101/2025.06.26.661135) also needs to be included. Based on such results, the authors should critically assess to what degree the added computational complexity provides substantial performance gains over simple baselines.
* The reliance on a single dataset for validation (two cell lines from the same paper, to be precise) is a major weakness and leaves it entirely open how well the method may generalize.
* The authors illustrate mean-collapse in Figure 1 based on correlations. However, I think MSE, or weighted MSE/weighted R2 with higher weight on DEG genes would be more appropriate for assessing the issue. Such metrics have been discussed in prior work (e.g., https://doi.org/10.1101/2025.10.20.683304).
* The authors evaluate their model on unseen perturbations, but it would also be valuable to assess how well the model generalizes to unseen cell lines, and potentially across different cell types and donors.
* The authors say that the subgraph is selected by differentiable node sampling based on the T threshold. However, they do not explain how this threshold is selected and how it affects the performance.
* The authors claim that, although biological knowledge graphs improve performance, existing methods typically use them as static structures. It would be useful to include an additional baseline in which AdaPert uses fixed subgraphs, allowing a more direct evaluation of the impact of the adaptive subgraph selection.
* The authors claim that the responses are sparse, yet the evaluation is primarily focused on DEG genes. It would be helpful to report similar metrics for non-DEG genes to support the claim that these genes are not substantially affected by the predictions.
* It would be helpful to explain why mean-collapse is compared only between AdaPert and TxPert.

---

> ### Author Rebuttal · Authors · 2026-03-31
>
> We sincerely thank the reviewer for the thorough and constructive evaluation, and for recognizing that AdaPert is "technically sound" with "well-designed experiments," that the adaptive subgraph learning is "clearly stated," and that our work addresses "an important aspect" of this "rapidly evolving field" with ideas "likely to motivate future work." We address each concern below.
>
> **[W1, W8, W9]** We conducted experiments to quantify KG dependency and validate the adaptive mechanism (Please refer to https://anonymous.4open.science/api/repo/rebuttal-AF11/file/index.html#nSvh_kg and #BXkW responses for full details). As shown in `Figure 3`, fixed 1-hop and 2-hop aggregation perform comparably, yet neither matches adaptive extraction. This is consistent with our finding in `Figure 9 in Appendix`, where ground-truth DEGs spread across distant hops rather than concentrating near the perturbation gene. And `Figure 4` validates this, high-attention node counts closely match DEG counts at each hop distance, confirming biologically grounded selection.
>
> [W2&W3] DEGs were identified using the Wilcoxon rank-sum test, followed by Benjamini-Hochberg FDR correction. Default threshold is FDR < 0.05 with |LFC| > 0.25. The expression change 0.1 in line 713 refers to the criterion of statistical analysis, not used in either the training or test step. We will clarify this in the revision. Sensitivity analysis varying FDR and LFC cutoff across three cell lines confirms that a moderate threshold (FDR<0.05, |LFC|>0.25) consistently performs best. Please refer to Reviewer #mrYu[W1&Q1] for full details.
>
> **[W4, W5, W7, Q1]** Please refer to https://anonymous.4open.science/api/repo/rebuttal-AF11/file/index.html#nSvh_comprehensive .
>
> **[W4]** We expanded baselines to include mean baselines and linear models with one-hot/GenePT embeddings (Nature Methods, 2025). Across four cell lines: (1) Simple baselines are competitive but inconsistent, and no simple baseline consistently ranks second. (2) AdaPert outperforms the best linear baseline by a substantial margin in every setting. (3) Overhead is modest: only sub-KG extraction and DEG-guidance loss added (1M vs 8.5M params).
>
> **[W5, W7, Q1]** We clarify the dataset confusion and expand evaluation: (1) Added HEPG2 (`Figure 2`) and JURKAT (`Figure 3`). AdaPert ranks first across all four cell lines on most metrics (`Figs 1-5`). (2) Under unseen cell line setting (Figure 5), Perturb Mean leads due to mean-aware construction. Among learned methods, AdaPert and Linear lead, both outperforming TxPert and foundation models. (3) Under Systema benchmark, AdaPert achieves the best Pearson-Delta and Centroid Accuracy across all five settings(`#mrYu#Q3`).
>
> **[W6, W10, W11, Q2, Q3]** Please refer to https://anonymous.4open.science/api/repo/rebuttal-AF11/file/index.html#nSvh_collapse .
> These five points center on mean-collapse. We address them with three analyses suggested by the reviewer to support the mean-collapse hypothesis[Q3].  (1) Rank-weighted DEG metrics confirm gains concentrate on perturbation-affected genes (**W6, Figure 1**). (2) Non-DEG metrics show no degradation (**W10, Figure 1**). (3) Stratified ablation reveals context extraction has the largest impact where sparsity is most severe (**Q2, Figure 2**). AdaPert's improvement concentrates on DEGs while non-DEG prediction remains intact, confirming the mitigation of the mean collapse effect.
>
> **[W11]** We use TxPert for comparison because it uses the knowledge graph on perturbation prediction, making it the most controlled comparison.
>
> **[Q2]** Thank you for this suggestion. We now report both mean and median, split by perturbation strength (`Figure 2`). The context module helps most at the extremes. Weak perturbations on directional metrics, strong perturbations on discrimination, showing that the adaptive components address the collapse problem where it matters, not across the board.
>
> **[Q4]** We thank the reviewer for this suggestion. We evaluated downstream utility through three analyses on K562, comparing AdaPert with TxPert and Linear-GenePT. Please refer to https://anonymous.4open.science/api/repo/rebuttal-AF11/file/index.html#nSvh_downstream.  **DEG identification** (`Figure 1`): AdaPert improves DEG recovery over TxPert and direction prediction. Full details in caption. **Pathway recovery** (`Figure 2`): GSEA on predicted profiles shows AdaPert recovers 20.0% of ground-truth significant pathways, with the highest direction accuracy. **Target prioritization** (`Figure 3`): Perturbations ranked by predicted pathway effect against ground truth. AdaPert achieves clear advantages on biologically relevant pathways, including Glycolysis and mTORC1 Signaling pathways. These results confirm that AdaPert's metric improvements translate into practical gains for DEG identification, pathway analysis, and perturbation-based target selection. We will add these results in revised manuscript.

---

> > ### Author Rebuttal · Reviewer_nSvh · 2026-04-03
> >
> > Thank you for the detailed responses. I appreciated the additional analyses and benchmarks. However, several points still require clarification:
> > 1. Figure captions of the additional results should be more descriptive and detailed to facilitate easier interpretation. For instance, in the 'KG_variations' section, it is not clear which datasets were used for the evaluation.
> > 2. A more comprehensive assessment of the results in the 'downstream_analysis' section would be appreciated. In particular, additional context regarding the tested cell line and the biological relevance of the enriched pathways would help highlight the significance of the results, especially for readers without biological background. I also suggest including results for all hallmarks rather then a selected subset.
> > 3. There is an inconsistently in section 'comprehensive_results' between Figure 5 and Table 5. If I understand correctly, they report the same metrics, but the Pearson delta for the linear model is missing on the plot, with the highest result reported in the table. Additionally, Table 5 refers to a method 'STAMP', I suspect it should be 'STATE'.
> > 4. The statement 'And Figure 4 validates this, high-attention node counts closely match DEG counts at each hop distance, confirming biologically grounded selection' seems to be an overstatement. Based on Figure 4, in particular at the optimal T=0.1, the number of selected DEG is rather low compared to the 'ground-truth' DEG counts.
> > 5. I also suggest including a limitations/discussion section in the revised submission to provide a clearer statement of the method’s contributions given the updated benchmarks.

---

> > > ### Author Response · Authors · 2026-04-05
> > >
> > > We thank the reviewer for the careful follow-up and the specific suggestions. Your comments helped us improve both the clarity and completeness of our results. We address each point below.
> > >
> > > **[1] Figure captions in KG_variations.** Thank you for pointing this out. We have updated the figure captions with detailed descriptions, including the dataset (K562), experimental conditions, and what each subplot shows. Please refer to `Figures 1–4` via https://anonymous.4open.science/api/repo/rebuttal-AF11/file/index.html#nSvh_kg.
> > >
> > > **[2] Biological context and full hallmark results.** Thank you for this excellent suggestion. Please refer to `Figure 3` (subset, first-round rebuttal) and `Figure 4` (complete 50 pathways with K562-relevant annotations) via https://anonymous.4open.science/api/repo/rebuttal-AF11/file/index.html#nSvh_downstream.
> > > (1) To assist readers from a computational background, we updated the captions in `Figures 1–4` and provided a biological context for the tested cell line. K562 is a well-characterized chronic myeloid leukemia cell line with constitutively active PI3K/AKT/mTOR signaling and metabolic reprogramming (Warburg effect), which makes it a useful benchmark for checking whether predicted perturbation effects are biologically meaningful. We also clarified why pathway-level evaluation matters here: it helps show that AdaPert captures biological functions (e.g., metabolic shifts) **rather than memorizing numerical expression patterns**.
> > > (2) Following your suggestion, we expanded the evaluation to **all 50 MSigDB Hallmark gene sets** in `Figure 4`. We find it reveals a clear biological trend: AdaPert consistently outperforms baselines on **K562-relevant pathways**`(highlighted in orange)`, with gains concentrated on core metabolic and oncogenic processes such as Heme Metabolism, Glycolysis, Cholesterol Homeostasis, and mTORC1 Signaling. Baselines only focus on general cellular processes (e.g., Apoptosis, G2-M Checkpoint), less specific to the erythroleukemia phenotype.
> > >
> > > **[3] Figure 5 / Table 5 update.** Sorry for the confusion, and please check the updated contents via https://anonymous.4open.science/api/repo/rebuttal-AF11/file/index.html#nSvh_comprehensive. We corrected both: (1) "STAMP" → "STATE" in Table 5 and (2) an HTML rendering error that displaced the Linear model's results (corrected Pearson Δ = 0.515). This update also prompted us to look more carefully at the cross-cell-line results. AdaPert is designed for unseen perturbation prediction within a given cellular context, where it consistently achieves the best performance in the first four datasets. The cross-cell-line setting poses a fundamentally different challenge: transferring across cellular contexts with **limited training diversity**. We find simpler baselines outperform deep learning methods, which is likely due to **extreme data skew** (87% of test perturbations have conserved cross-cell-line effects) and a low-context regime (only 4 training cell lines) where **complex models overfit** cell-line-specific patterns. Extending AdaPert to handle cross-cell-line transfer is a promising future direction, which we will discuss in the Limitations section.
> > >
> > > **[4] Tone down on attention-DEG alignment**. We appreciate this feedback and agree that "closely match" overstates the observation at T=0.1. We have revised the caption to state that attention-based filtering reduces the neighborhood to the scale of DEG counts, providing a "reasonable approximation of perturbation-relevant context" rather than claiming a close quantitative match. The updated caption is available in `Figure 4` via https://anonymous.4open.science/api/repo/rebuttal-AF11/file/index.html#nSvh_kg.
> > >
> > > **[5] Limitations/Discussion section.** We will add a `Discussion and Limitations section` in the revised manuscript. As you suggested, we will discuss how perturbation-specific contexts learned through DEG-aware supervision directly target the mean collapse problem and capture biologically meaningful signals for unseen perturbations more effectively than static graph methods and foundation models. Second, we will discuss current limitations and future directions, including cross-cell-line transfer and the efficient usage of knowledge graphs.
> > >
> > > We are grateful for the reviewer's engagement across both rounds of review. The constructive suggestions to add biological context, expand to all 50 hallmark pathways, and clarify cross-cell-line results have meaningfully strengthened our manuscript. All updates will be reflected in the revised version.

---

### Official Review · Reviewer_mrYu · 2026-03-12

**Soundness:** 3
**Presentation:** 4
**Significance:** 3
**Originality:** 2
**Overall Recommendation:** 5
**Confidence:** 3

**Summary:**

This paper presents ADAPERT, a framework for transcriptional response prediction to perturbations that focuses on differentially expressed genes (DEGs) and uses adaptive learning to rely on perturbation-specific knowledge subgraphs. The authors build on extensive recent literature showing that many existing models used to this end look good on correlation-based metrics by simply replicating the expression mean on the unperturbed data. Since most genes are not affected normally by the perturbation at hand, predictions look good while not capturing anything about the underlying biology. To mitigate this problem, ADAPERT is trained to reconstruct the perturbed expression vectors with three losses:
1) A regular MSE on the full expression vector (Lrecon)
2) A Huber loss on non-DEGs, that penalises large deviations quadratically but small deviations linearly (to account for plain noise in the data), and
3) a response-aware alignment loss that encourages the extracted subgraph representation to encode perturbation specific signals.
The authors comprehensively show that ADAPERT beats the competition in several benchmarks focusing on two metrics: plain correlation (which can mask mean collapse) and perturbation discriminative score (PDS).

**Compliance With Llm Reviewing Policy:**

Affirmed.

**Final Justification:**

All my initial concerns were thoroughly addressed. I recommend to accept the manuscript.

**Key Questions For Authors:**

- Regarding DEGs: how dependent is ADAPERT on the statistical method used to distinguish between DEGs and non-DEGs? A comprehensive evaluation of significance thresholds, fold changes, and algorithms used would help assess how robust the presented method is to this critical step.
- Would it be possible to evaluate ADAPERT in the SYSTEMA framework? I believe this could be a unique opportunity to see the method shine.

**Limitations:**

I did not find a particular discussion on weaknesses or limitations.

**Strengths And Weaknesses:**

Strong points:
- Overall, the paper consistently shows ADAPERT improves performance over existing baselines
- The writing and presentation of the manuscript are excellent
- Ablation studies on the different components of the presented model are quite convincing
- The distingction between DEGs and non-DEGs during training is original (to the best of my knowledge)

Weak points:
- While original, the reliance on distinguishing between DEGs and non-DEGs makes the paper hinge on external statistical methods
- While as I said the presentation is excelent overall, some clarity on the adaptive learning step would be appreciated
- Maybe my main concern, there are recent baselines and frameworks tackling this problem that were neither discussed nor used in the current version of the manuscript (see STSTEMA --Viñas Torné et al, Nature biotechnology 2025).

---

> ### Author Rebuttal · Authors · 2026-03-31
>
> We sincerely thank the reviewer for the positive assessment and for recognizing that AdaPert "consistently shows improvement over existing baselines," that the "writing and presentation are excellent," and that the DEG/non-DEG distinction during training is "original." We address each remaining concern below.
>
> **[W1&Q1]** We thank the reviewer for this important suggestion. We conducted a comprehensive sensitivity analysis by varying FDR threshold (p-value) \{0.01, 0.05\} and log-fold-change cutoff \{none, |LFC|>0.25, |LFC|>0.50\}, yielding 6 combinations. AdaPert was retrained from scratch under each setting across three cell lines (K562, JURKAT, RPE1) and evaluated on all metrics in https://anonymous.4open.science/api/repo/rebuttal-AF11/file/index.html#mrYu_DEG.
> Despite DEG counts varying within a dataset (e.g., K562: 229–641, JURKAT: 63–222), performance remains remarkably stable: Pearson-Delta varies by <1.3% in K562, <2.1% in JURKAT, and <0.7% in RPE1, with similar stability across MAE, Discrimination, Overlap, and DE-Spearman. Notably, a moderate threshold (FDR<0.05, |LFC|>0.25) consistently performs best, balancing signal quality with sufficient DEG coverage for the non-DEG regularization to be effective. These results confirm that AdaPert's DEG-aware loss acts as a **soft regularizer** rather than a hard constraint, approximate DEG labels are sufficient, and provide standard thresholds without extensive tuning.
>
> **[W2]** We thank the reviewer for this suggestion. We clarify the adaptive learning mechanism at both training and inference stages, and provide a detailed inference diagram in https://anonymous.4open.science/api/repo/rebuttal-AF11/file/index.html#mrYu_adaptive_learning. **During training**, for each perturbation p, DEG labels `D(p)` are obtained via statistical testing between control and perturbed cell populations, reflecting statistically grounded, perturbation-specific biological signals. These labels drive two forms of adaptation: (1) progressively refining which subgraph structures correspond to biologically meaningful responses across training iterations; (2) the adaptive loss adjusts its scope per perturbation based on each perturbation's DEG/non-DEG partition, automatically adapting to different signal-to-noise profiles.
> **In the inference stage**, for an unseen gene `p`, a GNN on the shared KG produces structural embeddings `h_v`, while an LM provides a semantic embedding `s_p`. Each node is scored by `sigma(W_c[h_v || \hat{s}_p])`, and nodes exceeding threshold `T` are selected to form `G_context`. The context representation `z_context = sum_{v in G_context} * alpha_v * h_v` is combined with the encoded control state `z_c` and `\hat{s}_p` as `[z_c, z_context, \hat{s}_p]` for decoding. Crucially, no DEG labels, no losses, and no perturbation-specific graph construction are needed in the inference stage. DEG labels serve solely as training-time supervision that guides transferable signal-noise separation, and this learned capacity generalizes to unseen perturbations at inference.
>
> **[W3&Q2]** We thank the reviewer for bringing SYSTEMA to our attention. To resolve your concerns, we expanded our evaluation to the SYSTEMA framework in https://anonymous.4open.science/api/repo/rebuttal-AF11/file/index.html#mrYu_systema. To this end, included additional baselines (Linear One-hot, Linear GenePT, scGPT, STATE) and two additional cell lines (HEPG2, JURKAT), as well as an unseen cell line setting (K562-cross). All the methods were evaluated within the SYSTEMA framework, reporting Pearson-Delta against both perturbation and control references, Pearson-Delta@20 (top-20 DE genes), and Centroid Accuracy. AdaPert consistently achieves the highest Pearson-Delta (perturbation reference) across all four single-cell-line benchmarks, outperforming the second-best method by +3.2% to +5.1%. On Centroid Accuracy, AdaPert also ranks first. In the cross-cell-line setting, AdaPert achieves P-Delta_pert = 0.783, outperforming the second-best (STATE) by +6.4%. Foundation models (scGPT: 0.047–0.086) and classical methods (CPA, scVI) show substantially lower scores, consistent with the mean-collapse phenomenon discussed in our paper. We will add a discussion of SYSTEMA in Related Work and include full benchmark results in the revised manuscript.
>
> **[Further improvements]** We have added a Limitations section discussing current scope assumptions, including DEG label dependency and knowledge graph coverage, along with directions for future work to address them.

---

> > ### Author Rebuttal · Reviewer_mrYu · 2026-04-05
> >
> > I thank the authors for thoroughly implementing my suggestions and addressing my concerns. The addition of SYSTEMA is highly appreciated in such a short time frame, and all results look convincing to me. I have adjusted my score.

---

> > > ### Author Response · Authors · 2026-04-05
> > >
> > > Thank you for taking the time to revisit our work and for the constructive suggestions. We are glad our rebuttal addressed your concerns. We will integrate the DEG sensitivity analysis, SYSTEMA benchmark, and a clearer description of the adaptive learning process during the rebuttal into the revised manuscript, along with a new Discussion section. Your suggestions helped us strengthen the paper.

---

### Decision · Program_Chairs · 2026-04-30

**Decision:**

Accept (regular)

**Comment:**

The paper addresses the mean-collapse problem in single-cell genetic perturbation prediction, where models achieve high correlation by predicting global average expression rather than perturbation-specific responses. The method learns perturbation-specific subgraphs from biological knowledge graphs and applies an adaptive training objective that separates DEG (differentially expressed gene) signals from non-DEG noise via a Huber-based non-DEG loss and a response-aware alignment loss. The proposal is well motivated and empirically validated, and demonstrates consistent improvements across multiple cell lines, baselines, and evaluation frameworks. The rebuttal results further strengthen the empirical contribution, and the remaining concerns are mostly addressable in the camera-ready. Therefore, I recommend acceptance.